# A Unified Knowledge Distillation Framework for Deep Directed Graphical Models

## Abstract

Knowledge distillation (KD) is a technique that transfers the knowledge from a large teacher network to a small student network. It has been widely applied to many different tasks, such as model compression and federated learning. However, the existing KD methods fail to generalize to general *deep directed graphical models (DGMs)* with arbitrary layers of random variables. We refer by *deep* DGMs to DGMs whose conditional distributions are *parameterized by deep neural networks*. In this work, we propose a novel unified knowledge distillation framework for deep DGMs on various applications. Specifically, we leverage the reparameterization trick to hide the intermediate latent variables, resulting in a compact DGM. Then we develop a surrogate distillation loss to reduce error accumulation through multiple layers of random variables. Moreover, we present the connections between our method and some existing knowledge distillation approaches. The proposed framework is evaluated on three applications: deep generative models compression, discriminative deep DGMs compression, and VAE continual learning. The results show that our distillation method outperforms the baselines in data-free compression of deep generative models, including variational autoencoder (VAE), variational recurrent neural networks (VRNN), and Helmholtz Machine (HM). Moreover, our method achieves good performance for discriminative deep DGMs compression. Finally, we also demonstrate that it significantly improves the continual learning performance of VAE.

## 1 Introduction

Knowledge distillation (KD) aims at transferring the knowledge of a large teacher model to a small student model, which tries to mimic the behavior of the teacher model to attain a competitive or superior performance Hinton et al. (2015); Gou et al. (2021). The goal of this work is to develop a *unified knowledge distillation (KD) framework* for *deep directed graphical models* (DGMs). Applications of the proposed framework include: (i) data-free deep generative models compression, (ii) discriminative deep DGMs compression, and (iii) continual learning with VAEs.

*Deep directed graphical models* (DGMs) refer to DGMs whose conditional distributions are parameterized by deep neural networks (DNNs), which is in contrast to the regular DGMs with tabular conditional probability. One good example is variational autoencoders (VAEs), whose posterior probability of latent variables is parameterized by DNNs. A general deep DGM may have complex structures, consisting of arbitrary number of input variables, target variables, and latent variables. Deep DGMs have been widely used in various applications, such as image generation Vahdat & Kautz (2020), text generation Bowman et al. (2015), and video prediction Wu et al. (2021).

This work is motivated by the growing popularity of recent over-parameterized deep DGMs with millions of parameters to improve their accuracy in various tasks. However, the large models are very computationally expensive. As a result, it is *not* practical to deploy them on resource-constrained edge devices, such as mobile phones and IoT systems Li et al. (2020). One possible solution to this problem is KD, which enables a smaller student model to approximate the performance of a large teacher. Recently, KD has been widely applied to many different tasks, such as model compression Hinton et al. (2015), continual learning Li & Hoiem (2017); Zhai et al. (2019), and federated learning Lin et al. (2020b); Li & Wang (2019). To our knowledge, the existing KD methods, however, are only applicable to some specific DGMs, including generative adversarial networks (GANs) Aguinaldo

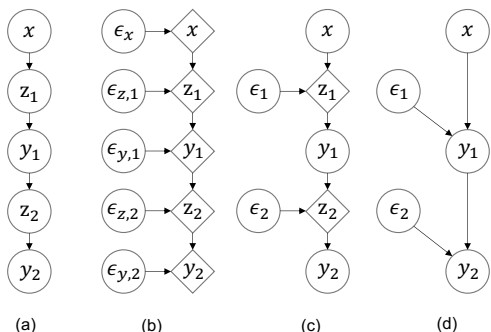 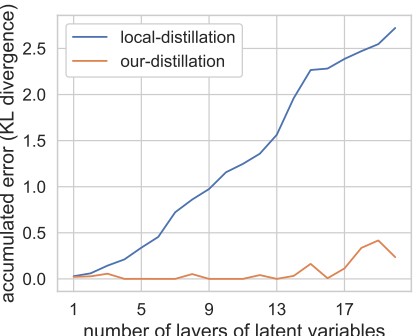

Figure 1: Illustration of DGM in four different forms. Diamonds are deterministic variables and circles are random variables. (a) Original form; (b) Auxiliary form; (c) Our semi-auxiliary form; (d) Compact semi-auxiliary form.

Figure 2: Toy example of accumulated error (KL divergence) between the teacher and student for local distillation and our method. Experimental settings are presented in the last paragraph in Section 3.2.

et al. (2019); Li et al. (2020), auto-regressive models in natural language processing (NLP) Lin et al. (2020a), and VQ-VAE Roy et al. (2018). *They fail to generalize to the general deep DGMs, especially to those with multiple latent variables or complex dependence structures*, as illustrated in Fig. 1.

Generalizing knowledge distillation to deep DGMs poses two major challenges. First, distillation by marginalizing all latent variables is generally intractable. Secondly, distilling each layer locally and independently may suffer from error accumulation, as shown Fig. 2. We can observe that the accumulated error (i.e., KL divergence) between the teacher and student grows linearly for local distillation. To address these challenges, we propose a novel unified knowledge distillation framework for deep DGMs. Specifically, we first adopt the reparameterization trick Kingma (2013); Kingma et al. (2014) to convert a DGM into a compact *semi-auxiliary form*. By *semi-auxiliary form*, we mean the latent variables, $z$, in both the student and teacher models are converted to deterministic variables with auxiliary variables, while the input variables and target variables remain unchanged, as shown in Fig. 1 (c). Note that different from the classical reparameterization for model training that requires continuous latent variables, ours can be applied to both continuous and discrete variables. Then a surrogate distillation loss is derived as a new objective of KD. To mitigate gradient vanishing, we further incorporate a latent distillation loss that penalizes the dissimilarity of latent variables between the teacher and student into our objective. Moreover, we illustrate the connections between our approach and some existing KD methods for specific DGMs. We demonstrate that our method is a proper generalization of these existing methods.

We apply our distillation method to three different applications: deep generative models compression Rezende et al. (2014); Chung et al. (2015), VAE continual learning Rao et al. (2019), and discriminative deep DGMs compression Alemi et al. (2016). For deep generative models compression, the student model distilled by our method in a data-free manner outperforms that trains from scratch and the other baselines. In addition to generative modeling, our method is able to compress *discriminative* deep DGMs with high accuracy. Finally, we also illustrate that it can better mitigate the catastrophic forgetting issue than the generative replay approach in continual learning.

In summary, our contributions include: 1) a new unified KD framework is proposed for general deep DGMs based on reparameterization trick, 2) we derive a novel distillation loss that combines the latent distillation loss and surrogate distillation loss to improve the performance of KD, 3) evaluation results on multiple benchmark datasets show that our approach can not only achieve high accuracy for deep DGMs compression but also improve the continual learning performance of VAEs.

## 2 PRELIMINARIES

**Directed Graphical Models (DGMs)** DGMs such as Bayesian Networks (Darwiche, 2009) are an expressive class of probabilistic graphical models (Koller & Friedman, 2009), in which the joint distribution is factorized into the product of many conditional distributions according to a directed acyclic graph (DAG) that captures variable conditional dependencies. In this work, we primarily

study knowledge distillation for deep DGMs, especially for those with complicated dependency structures.

For deep DGMs, we are interested in modeling the conditional distribution $p_\theta(\boldsymbol{y}, \boldsymbol{z}|\boldsymbol{x})$ for *target variables* $\boldsymbol{y}$ and *latent variables* $\boldsymbol{z}$ given *input variables* $\boldsymbol{x}$, parameterized by $\theta$. Specifically, when there is no input variables, i.e., $\boldsymbol{x} = \emptyset$, we actually model the joint distribution $p_\theta(\boldsymbol{y}, \boldsymbol{z})$. Let $Pa(\cdot)$ denote the parent random variables of a certain variable which is defined by the DAG of a DGM. Then the conditional distribution $p_\theta(\boldsymbol{y}, \boldsymbol{z}|\boldsymbol{x})$ has its factorized form as follows,

$$p_\theta(\boldsymbol{y}, \boldsymbol{z}|\boldsymbol{x}) = \prod_j p_\theta(\boldsymbol{y}_j|Pa(\boldsymbol{y}_j), \boldsymbol{x}) \prod_i p_\theta(\boldsymbol{z}_i|Pa(\boldsymbol{z}_i), \boldsymbol{x}), \quad (1)$$

where $\boldsymbol{y}_j$ denotes the $j$th target variable in $\boldsymbol{y}$, $\boldsymbol{z}_i$ denotes the $i$th latent variable in $\boldsymbol{z}$. Without loss of generality, we assume for two variables $\boldsymbol{y}_i$ and $\boldsymbol{y}_j$, if $\boldsymbol{y}_j$ is an ancestor of $\boldsymbol{y}_i$, then it holds that $i > j$.

**Knowledge Distillation (KD)**   KD aims to transfer the knowledge of a large teacher model to a smaller student model. One commonly used vanilla distillation method is to encourage the student to mimic the output of the teacher model Hinton et al. (2015). Given an empirical distribution of the training data $p_{data}(\boldsymbol{x})$, its distillation loss is an expected dissimilarity measure between the output of the student and that of the teacher. A general form of distillation loss is given by

$$\mathcal{L}_{kd} = \mathbb{E}_{p_{data}(\boldsymbol{x})} \left[ d(p_\phi(\boldsymbol{y}|\boldsymbol{x}), p_\theta(\boldsymbol{y}|\boldsymbol{x})) \right], \quad (2)$$

where $p_\phi(\boldsymbol{y}|\boldsymbol{x})$ and $p_\theta(\boldsymbol{y}|\boldsymbol{x})$ denote the output conditional distribution of the teacher and student models, respectively. $\phi$ and $\theta$ denote their corresponding parameters. $d(\cdot, \cdot)$ is a dissimilarity measure between two probability distributions. Kullback-Leibler (KL) divergence (Hinton et al., 2015; Chen et al., 2017), for example, is one of some typical choices.

The above Eq. (2) and its extended version have been applied to different DGMs, such as vanilla neural networks Chen et al. (2017), GANs Aguinaldo et al. (2019), and fully-visible auto-regressive models (e.g., Transformer) Kim & Rush (2016); Lin et al. (2020a). Take fully-visible auto-regressive models as an example. When we set $d(\cdot, \cdot)$ to KL divergence, Eq. (2) can be factorized as

$$\mathcal{L}_{kd} = \mathbb{E}_{p_{data}(\boldsymbol{x})} \left[ \sum_j \mathbb{E}_{p_\phi(\boldsymbol{y}_{<j}|\boldsymbol{x})} \left[ KL(p_\phi(\boldsymbol{y}_j|\boldsymbol{y}_{<j}, \boldsymbol{x}) \, \| \, p_\theta(\boldsymbol{y}_j|\boldsymbol{y}_{<j}, \boldsymbol{x})]) \right] \right]. \quad (3)$$

A tractable estimation to $\mathcal{L}_{kd}$ above can be obtained by Monte Carlo method. It can also be viewed as distilling each conditional distribution in a local and independent manner, as shown in Fig. 3 (d).

However, to our best knowledge, the existing KD approaches are only designed for some specific DGMs. They fail to be applied to a general DGM with multiple latent variables or complicated dependence structures. Hence, the question is, how can we distill the knowledge from the teacher to the student given a general DGM structure?

Intuitively, there are two naive methods: (i) *marginalized distillation*: it marginalizes all latent variables to get $p(\boldsymbol{y}|\boldsymbol{x}) = \int p(\boldsymbol{y}, \boldsymbol{z}|\boldsymbol{x})\mathrm{d}\boldsymbol{z}$. However, the integration $\int p(\boldsymbol{y}, \boldsymbol{z}|\boldsymbol{x})\, d\boldsymbol{z}$ is generally intractable and thus the loss as in Eq. (2) is also intractable. (ii) *local distillation*: it treats latent variables $\boldsymbol{z}$ equally as target variables and then distills each conditional distributions for both $\boldsymbol{z}$ and $\boldsymbol{y}$ locally and independently, as shown in Fig. 3(d). The local distillation may suffer from error accumulation through multiple layers if each conditional distribution in the student slightly deviates from the teacher. Fig. 2 shows a toy example that the accumulated error of local distillation, i.e., the KL-divergence between teacher and student, increases linearly as the number of layers of latent variables rises. For detailed discussions, please refer to Appendix A.

**Reparameterization Trick**   Reparameterization trick Kingma (2013); Kingma et al. (2014), also called the auxiliary form of a DGM, is originally proposed to backpropagate through a random node that is not differentiable during training. Its basic idea is introduced as follows. Given a conditional distribution of random variable $\boldsymbol{z}_i$ in a DGM, $p(\boldsymbol{z}_i|Pa(\boldsymbol{z}_i), \boldsymbol{x})$, we convert it to a deterministic variable by adding an *auxiliary variable* $\boldsymbol{\epsilon}_i$ to its dependence, as shown in Figure 1 (c). Here $\boldsymbol{\epsilon}_i$ is a root node of the DGM with an independent marginal distribution of $p(\boldsymbol{\epsilon}_i)$. By choosing appropriate $p(\boldsymbol{\epsilon}_i)$ and deterministic transformation $g(\cdot)$ Kingma (2013), we can have $\boldsymbol{z}_i = g(Pa(\boldsymbol{z}_i), \boldsymbol{x}, \boldsymbol{\epsilon}_i)$, where $\boldsymbol{z}_i$ is determined by its parent variables $Pa(\boldsymbol{z}_i)$, input variables $\boldsymbol{x}$, and the corresponding auxiliary variable $\boldsymbol{\epsilon}_i$. $\boldsymbol{\epsilon}_i$ serves as the source of stochasticity of $\boldsymbol{z}_i$.

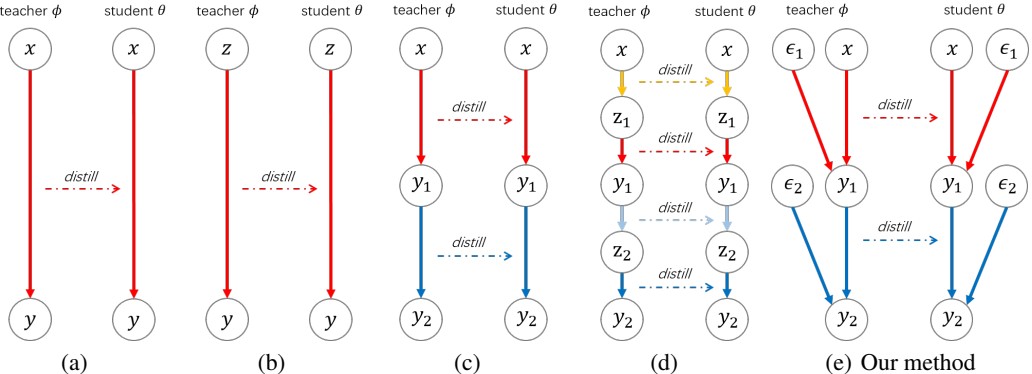

Figure 3: Illustration of different distillation methods. Each pair of conditional distributions marked with same color represents an independent distillation component. (a) Distillation on vanilla neural network. (b) GAN distillation. (c) Distillation on a 2-layer fully-visible auto-regressive DGM. (d) Local distillation on the original DGM. (e) Our distillation method with semi-auxiliary form of DGM.

In this work, note that we do not primarily use reparameterization trick for model training. Rather, we leverage it to convert the latent variables $z$ in DGMs to deterministic variables so that we can effectively distill knowledge from a compact form of DGM. Note that different from the classical reparameterization for model training that requires continuous latent variables, ours can be applied to a wider range of variables $z$, including both continuous and discrete variables. Besides, the transformation function $g(\cdot)$ Kingma & Welling (2013) in our framework can be either differentiable or non-differentiable. Hence, our method can be applied to much more DGMs than the classical one. Below, we will elaborate this general idea in more detail.

## 3 MODELING

In this section, we first introduce the semi-auxiliary form of DGMs using reparameterization trick. Then we propose a new surrogate loss function and latent distillation loss for our KD method.

### 3.1 SEMI-AUXILIARY FORM

As discussed in Section 2, the two naive methods, marginalized distillation and local distillation, do not work well due to intractable distillation loss or error accumulation. In order to address these issues, we propose a novel idea that converts DGM to its *semi-auxiliary form* based on reparameterization trick. Specifically, we convert all the latent variables, $z$, in both the teacher and student to deterministic variables with auxiliary variables, while keeping target variables $y$ and input variables $x$ unchanged. This is because our ultimate goal is to encourage student to mimic the output (target variable) of the teacher based on input variables. Hence, we can omit the deterministic (latent) variables in a DGM, yielding a compact semi-auxiliary form that only consists of target variables, input variables and auxiliary variables. In this way, each target variable has a tractable and direct dependence on input variables or prior target variables, as shown in Fig. 1 (c).

Fig. 1 illustrates a toy example of converting DGM to its semi-auxiliary form. Fig. 1 (a) is the original form of a DGM, and its corresponding auxiliary form is shown in Fig. 1 (b). In this paper, we only assign auxiliary variables to latent variables $z$, as shown in Fig. 1 (c). We can observe from it that latent variables $z_1$ and $z_2$ are deterministic when their ancestors are given. We thus can omit the deterministic variables, leading to a compact semi-auxiliary form, as shown in Fig. 1 (d).

### 3.2 SURROGATE DISTILLATION LOSS

After obtaining the semi-auxiliary form of a DGM in Fig. 1 (d), our distillation loss becomes the dissimilarity between $p_\phi(\boldsymbol{y}|\boldsymbol{\epsilon}, \boldsymbol{x})$ and $p_\theta(\boldsymbol{y}|\boldsymbol{\epsilon}, \boldsymbol{x})$. We call it *surrogate distillation loss*. Our goal is to minimize the surrogate distillation loss w.r.t. student's parameters $\theta$ below.

$$\mathcal{L}_{sd} = \mathbb{E}_{p_\phi(\boldsymbol{\epsilon})p_{data}(\boldsymbol{x})}\left[d(p_\phi(\boldsymbol{y}|\boldsymbol{\epsilon}, \boldsymbol{x}), p_\theta(\boldsymbol{y}|\boldsymbol{\epsilon}, \boldsymbol{x}))\right], \tag{4}$$

where $\epsilon$ denotes a set of auxiliary variables $\epsilon$. The expectation of dissimilarity is taken over both empirical data distribution $p_{data}(\boldsymbol{x})$ and auxiliary variable distribution $p_\phi(\boldsymbol{\epsilon})$. The expectation can be estimated using Monte Carlo method. Note that $p_\phi(\boldsymbol{\epsilon})$ is generally chosen to be simple and fixed distribution with no parameters, such as unit Gaussian or standard uniform distribution. Thus, it implies that teacher's $p_\phi(\boldsymbol{\epsilon})$ is equivalent to student's $p_\theta(\boldsymbol{\epsilon})$, so there is no need to distill $p_\phi(\boldsymbol{\epsilon})$ to $p_\theta(\boldsymbol{\epsilon})$. An illustration is given in Fig. 3(e).

**Proposition 3.1.** *The surrogate distillation loss as defined in Eq.* (4) *is an upper bound of the distillation loss as defined in Eq.* (2) *when the dissimilarity measure is chosen to be KL divergence.*

We provide the detailed proof in Appendix B.

Next, we discuss the advantages of our method over the two naive methods mentioned above. Firstly, the proposed method bypasses the intractable computation in marginalized distillation. While marginalized distillation measures $p(\boldsymbol{y}|\boldsymbol{x})$ which is intractable in general, we instead measure $p(\boldsymbol{y}|\boldsymbol{\epsilon}, \boldsymbol{x})$ which is easily tractable by function composition with no need of integral or sum operation. Here $p(\boldsymbol{y}|\boldsymbol{\epsilon}, \boldsymbol{x})$ is tractable because $p(\boldsymbol{y}|\boldsymbol{\epsilon}, \boldsymbol{x}) = \prod_i p(\boldsymbol{y}_i|\boldsymbol{\epsilon}_{\leq i}, \boldsymbol{y}_{<i}, \boldsymbol{x})$.

Secondly, our method can make the DGMs shallower than local distillation, mitigating error accumulation through multiple layers of latent variables. As illustrated in Fig. 3(d) and 3(e), we can see our method only constraints the target variables $\boldsymbol{y}$ in the student while local distillation constrains both latent variables $\boldsymbol{z}$ and target variables $\boldsymbol{y}$. As a result, local distillation may suffer from error accumulation issue. Fig. 2 shows a toy example of comparing the accumulated error, i.e., KL divergence between the teacher and student, for our method and local distillation. In this illustrative experiment, we let student mimic the output distribution of the teacher with $L$-layers latent variables for $L \in 1, \ldots, 20$. Each layer of the teacher follows the Gaussian distribution $p(\boldsymbol{z}_{i+1}|\boldsymbol{z}_i) = \mathcal{N}(\boldsymbol{\mu}(\boldsymbol{z}_i), 0.01\boldsymbol{I})$, where $\boldsymbol{\mu}(\boldsymbol{z}_i)$ is $\boldsymbol{z}_i^{1.1}$ for $\boldsymbol{z}_i \geq 0$ and $-(-\boldsymbol{z}_i)^{1.1}$ for $\boldsymbol{z}_i < 0$. $p(\boldsymbol{z}_1)$ is a uniform distribution $U[-1, 1]$. The student is parameterized by neural networks with proper residual structure He et al. (2016). Then *density ratio estimation* Nguyen et al. (2010); Sugiyama et al. (2012) is used to measure the KL divergence between the teacher and student. We can observe from Fig. 2 that the accumulated error (KL divergence) grows linearly w.r.t the number of layers for local distillation because each layer of the student deviates from the teacher to an extent. In contrast, the accumulated error (KL divergence) of our method increases slowly. Furthermore, we show that our method can still mitigate error accumulation issue for a DGM whose latent variables are discrete in Appendix I.

### 3.3 LATENT DISTILLATION LOSS

The surrogate distillation loss in Eq.(4) can sufficiently achieve a satisfactory performance for knowledge distillation. Nevertheless, there are still two limitations of the proposed method for some special DGMs. Firstly, it fails to back-propagate the gradients when there exist discrete latent variables. Secondly, it might suffer from gradient vanishing when the network structure of latent variables is very deep and complex.

To deal with these issues, we propose to penalize the dissimilarity of latent variables $\boldsymbol{z}$ in the teacher and student model. The resulting latent distillation loss for latent variables $\boldsymbol{z}_i$ is given by

$$\begin{aligned}
\tilde{\mathcal{L}}_{z,i} &= \mathbb{E}_{p_\phi(\boldsymbol{\epsilon})p_{data}(\boldsymbol{x})}\left[d\left[p_\phi(\boldsymbol{z}_i|\boldsymbol{\epsilon}, \boldsymbol{x}), p_\theta(\boldsymbol{z}_i|\boldsymbol{\epsilon}, \boldsymbol{x})\right]\right] \\
&= \mathbb{E}_{p_\phi(\boldsymbol{\epsilon})p_{data}(\boldsymbol{x})}\left[d(p_\phi(\boldsymbol{z}_i|\boldsymbol{\epsilon}_{\leq i}, \boldsymbol{x}), p_\theta(\boldsymbol{z}_i|\boldsymbol{\epsilon}_{\leq i}, \boldsymbol{x}))\right],
\end{aligned} \tag{5}$$

where $\boldsymbol{\epsilon}_{\leq i}$ is a set of all the ancestral auxiliary variables of $\boldsymbol{z}_i$. $\boldsymbol{z}_i$ is deterministic when $\boldsymbol{\epsilon}_{\leq i}$ and $\boldsymbol{x}$ are given. In order to better penalize the dissimilarity of latent variables $\boldsymbol{z}_i$, one good choice is to (only) convert the current $\boldsymbol{z}_i$ back to its original form by removing $\boldsymbol{\epsilon}_i$ from its dependence. Then we have the following latent distillation loss

$$\mathcal{L}_{z,i} = \mathbb{E}_{p_\phi(\boldsymbol{\epsilon})p_{data}(\boldsymbol{x})}\left[d(p_\phi(\boldsymbol{z}_i|\boldsymbol{\epsilon}_{<i}, \boldsymbol{x}), p_\theta(\boldsymbol{z}_i|\boldsymbol{\epsilon}_{<i}, \boldsymbol{x}))\right]. \tag{6}$$

The proposed latent distillation loss can benefit our optimization process. When latent variables are continuous, the latent distillation loss may provide shallower and supplementary supervisory signals to hasten the convergence. When latent variables are discrete, it can deal with the back-propagation cutting off problem.

### 3.4 FINAL TARGET-FREE LOSS

Combining the above surrogate distillation loss with latent distillation loss, our final distillation loss is given by

$$\mathcal{L}_{our} = \mathcal{L}_{sd} + \lambda \sum_i \mathcal{L}_{z,i}, \tag{7}$$

where $\lambda$ is a hyper-parameter that controls the importance of latent distillation loss. Similar to Hinton et al. (2015), our loss in Eq. (7) is a *target-free* distillation loss, which means target data is not required for calculating it. Eq. (7) can also be applied to DGMs with no input variables (i.e. $\boldsymbol{x} = \emptyset$). In this case, Eq. (7) can be computed in a completely *data-free* manner. We summarize our KD method for a general DGM in Algorithm 1 in Appendix C.

### 3.5 CONNECTIONS TO OTHER DISTILLATION METHODS

We present the connections between our method and some existing knowledge distillation methods. **Vanilla KD and Sequence-Level KD.** When there is no latent variable in a DGM, our distillation method in Eq. (4) is naturally reduced to Eq. (2) by removing the dependence on auxiliary variables $\epsilon$, which is a typical vanilla KD Hinton et al. (2015). Also, when the DGM is a fully visible auto-regressive model, by removing dependence on $\epsilon$ and letting $d(\cdot, \cdot)$ be KL divergence, Eq. (4) can be reduced to Eq. (3), which is the Monte Carlo approximation of intractable sequence-level distillation loss Kim & Rush (2016). Please note that further simplification has been made in Kim & Rush (2016) to enhance the practicability.

**Feature Based KD.** Feature based knowledge distillation uses the intermediary representations of a teacher network to supervise a student network Romero et al. (2014). When the teacher and student share the same size of latent features, the feature distillation loss can be written as

$$\mathcal{L}_f = r_f(f_\phi(\boldsymbol{x}), f_\theta(\boldsymbol{x})), \tag{8}$$

where $f_\phi(\boldsymbol{x})$ and $f_\theta(\boldsymbol{x})$ are intermediary deterministic features of the teacher and student, respectively. $r_f(\cdot, \cdot)$ is a distance between two vectorized feature maps. A vanilla neural network with multiple intermediate features can be viewed as a DGM with multiple deterministic latent variables. Deterministic variables can be viewed as following the degenerate distributions. In general, for $p \geq 1$, p-Wasserstein distance $(\inf \mathbb{E}[r(\boldsymbol{a}_1, \boldsymbol{a}_2)^p])^{\frac{1}{p}}$ between two degenerate distributions located at $\mathbf{a}_1$ and $\mathbf{a}_2$ is equivalent to $r(\mathbf{a}_1, \mathbf{a}_2)$. Thus, by viewing the intermediate features as latent variables following degenerate distributions, and choosing $d(\cdot, \cdot)$ in our latent distillation loss $\mathcal{L}_{z,i}$ as Wasserstein distance $(\inf \mathbb{E}[r_f(\boldsymbol{z}_\phi, \boldsymbol{z}_\theta)^p])^{\frac{1}{p}}$ , our latent distillation loss is reduced to feature distillation loss in Eq. (8).

**GAN Distillation.** GAN distillation in Li et al. (2020); Aguinaldo et al. (2019) also incorporates the idea of feature distillation into their model, which is given by

$$\mathcal{L}_{gan} = r_o(G_\phi(\boldsymbol{z}), G_\theta(\boldsymbol{z})) + r_f(f_\phi(\boldsymbol{z}), f_\theta(\boldsymbol{z})). \tag{9}$$

The first term above is the output distillation loss of the generator and the second one is the intermediary feature distillation loss. Similar to feature distillation loss, by viewing the intermediate features and generator output as latent variables following degenerate distributions, and choosing $d(\cdot, \cdot)$ in $\mathcal{L}_{sd}$ and $\mathcal{L}_{z,i}$ as $p$-Wasserstein distance $(\inf \mathbb{E}[r_o(\boldsymbol{y}_\phi, \boldsymbol{y}_\theta)^p])^{\frac{1}{p}}$ and $(\inf \mathbb{E}[r_f(\boldsymbol{z}_\phi, \boldsymbol{z}_\theta)^p])^{\frac{1}{p}}$ respectively, our final distillation loss in Eq. (7) can be reduced to GAN distillation loss as well.

### 3.6 APPLICATIONS

We evaluate the performance of our method in three applications below.

- **Data-Free Deep Generative Models Compression.** We first apply our method to compress deep generative models in a data-free manner. We conduct experiments on three representative models: VAE, VRNN, and Helmholtz Machine (HM) Chung et al. (2015); Saul et al. (1996). Recent studies show that VAEs generate better with over-parametrization Vahdat & Kautz (2020). However, it is challenging to deploy them to edge devices with limited computing resources. In this work, we apply our method to compress a large 5-layer hierarchical VAE model Sønderby et al. (2016) to smaller models, as illustrated in Fig. 6(a) in Appendix D. Then it is applied to compress a sequence generative model with many latent variables, VRNN. Finally, we adopt our method to

compress the HM model with discrete variables. The classical HM only has one target variable and each layer is parametrized by a linear transformation. In order to demonstrate the applicability of our method, we extend it to a 5-layer deep HM with two targets, as shown in Fig. 6(d).

- **VAE Continual Learning.** In addition to model compression, we evaluate the performance of our method on VAE continual learning. Our goal is to model a new distribution while retaining the ability of modeling a learned old distribution without access to the old dataset. Prior work on VAE continual learning Ye & Bors (2020); Ramapuram et al. (2020) mainly resort to generative replay strategy, in which a set of samples generated from VAE model learned on old datasets $p'_\theta(\boldsymbol{y})$ are reused when learning a new dataset. In fact, our experiments will show that generative replay is inferior to our distillation method for VAE continual learning because of the blurry nature of VAE generation.

- **Discriminative Deep DGMs Compression.** Finally, we apply the proposed method to compress a 2-layer discriminative DGM in Alemi et al. (2016), as shown in Fig. 6(e). Incorporating our distillation loss into the original training loss, we have the following loss function.

$$\mathcal{L}_{joint} = (1 - \alpha)\mathcal{L}_{train} + \alpha\mathcal{L}_{our}. \tag{10}$$

## 4 EXPERIMENTS

In this section, we evaluate the performance of our method on three applications mentioned in Section 3.6. We carry out extensive experiments on seven benchmark datasets: Old Faithful Geyser Härdle et al. (1991), IAM online handwriting Liwicki & Bunke (2005), SVHN Netzer et al. (2011), CIFAR10 Krizhevsky et al. (2009), CelebA Liu et al. (2015), and Fashion-MNIST Sohn et al. (2015). The detailed data splitting for training and testing is described in Appendix E. In addition, we present the detailed model configurations and hyperparameter settings in Appendix F. Source code will be publicly available upon publication.

### 4.1 EVALUATION ON DATA-FREE DEEP GENERATIVE MODELS COMPRESSION

We first apply our distillation method to data-free compression of deep generative models, including VAE, RVNN, and HM. We compare the performance of the student model using our method and two baselines: training from scratch and local distillation Achille et al. (2018). Frechet inception distance (FID) Heusel et al. (2017) is adopted to evaluate their generative performance.

**Data-Free Hierarchical VAE Compression.** We compress a large 5-layer hierarchical VAE Sønderby et al. (2016) to a smaller model. Experiments are conducted on three benchmark datasets: SVHN, CIFAR10, and CelebA. Fig. 4 illustrates the comparison of FID for different methods. We can see that our method consistently outperforms the baselines for different student model sizes. Importantly, our data-free distillation method outperforms student trained from scratch. This is because directly optimizing a capacity-limited student VAE may not learn a decent model that can generate high-fidelity samples. Conversely, our method can help student VAE learn better performance from the teacher.

To demonstrate the similarity of the teacher and student, we further compare the FID between samples generated from the student and teacher for different methods, as shown in Fig. 7 in Appendix G. We observe that our distillation method helps the student better approximate the teacher's performance. We also demonstrate that our method is not very sensitive to the hyper-parameter $\lambda$. Please refer to Appendix G.3 for more details.

**Data-free VRNN Compression.** We then adopt our distillation method to compress a deep sequence generative model, VRNN. Fig 5 shows the generated strokes by our method and the baselines. It can be observed that our method can generate more readable and clearer strokes than the baselines. The student VRNN distilled by our method in Fig 5 (c) has comparable generative performance to that of the teacher in Fig 5 (b). Hence, we can conclude that the proposed method is able to achieve good performance for data-free VRNN compression.

**Data-Free HM Compression.** In addition, our method is applied to compress Helmholtz Machine (HM) with *discrete latent variables*. The experiment results, as shown in Fig. 9 in Appendix H, show that it can still achieve good performance as the teacher for HM compression. For more details, please refer to the experimental results in Appendix H.

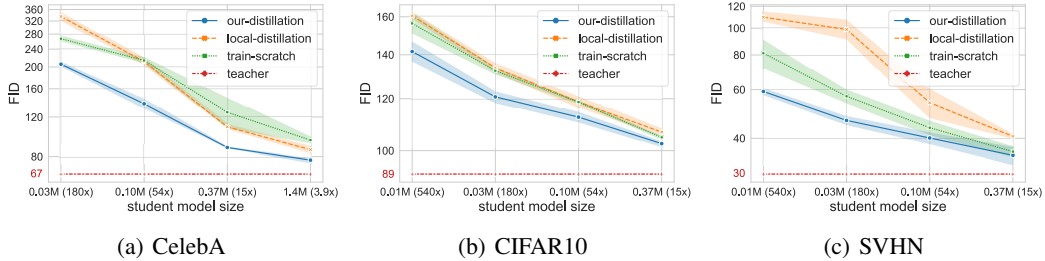

Figure 4: Comparison of FID for different methods in data-free VAE compression on three datasets: CelebA, CIFAR10, and SVHN. We compute the FID between the generated samples by student and ground truth averaged over 3 random seeds. Teacher model size is fixed to 5.4M parameters.

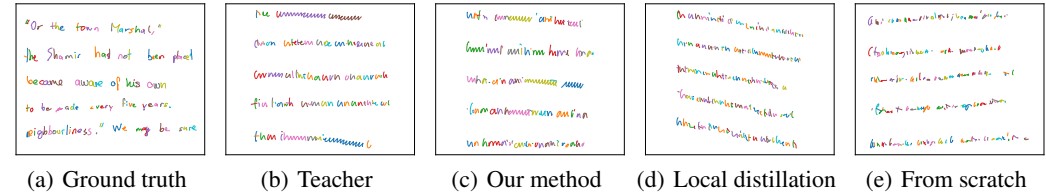

Figure 5: Strokes generated by VRNNs. Different strokes are represented with different colors. (a) Ground truth dataset. (b) Teacher VRNN (12M parameters) trained from scratch. (c) Student VRNN (2.4M parameters, 5.0 times smaller) distilled by our method. (d) Student VRNN distilled using local distillation. (e) Student VRNN trained from scratch.

## 4.2 EVALUATION ON VAE CONTINUAL LEARNING

We also evaluate the performance of the proposed method on VAE continual learning using CelebA dataset. For each experiment, one of forty ground-truth attributes is selected. Based on this attribute, we divide the whole dataset into two parts for continual learning. Specifically, we first learn to model one part of images, then we learn to model the other part while trying to retain the ability of modeling the first part. We repeat this experiment for four different attributes. The single input variable x in this model is a binary variable used to indicate which part the generated images belong to.

We compare our method with local distillation and other two generative replay approaches: CURL Rao et al. (2019)) and LGM Ramapuram et al. (2020). Table 1 illustrates the comparison of FID for different methods. We can observe from Columns Old that the increased FID of our method is much smaller than that of baselines, which means our method suffers from the least catastrophic forgetting. Besides, we can see from Columns New that our method achieves comparable FID to LGM and CURL on new distribution, because these methods put only necessary constraints on the model to prevent catastrophic forgetting. However, local distillation does not work well on new distributions because of its redundant constraints, leaving little free space for learning new distributions.

Table 1: FID of different methods after learning new distributions on CelebA dataset. Columns Old show FID between the generated images and real images on old distribution. A(+B) denotes that FID is increased by B to A after learning new distributions. Columns New illustrate FID between generated images and real images on new distributions. FID and (+B): lower is better.

| Methods/FID | No Eyeglasses → Eyeglasses | | Female → Male | | No Beard → Beard | | Not Smiling → Smiling | |
|---|---|---|---|---|---|---|---|---|
| | Old | New | Old | New | Old | New | Old | New |
| Our method | **79.0(+1.2)** | 79.8 | **85.1(+2.8)** | 69.3 | **80.0(+0.4)** | 79.8 | **90.1(+0.7)** | 78.1 |
| Local distillation | 178.7(+100.9) | 122.2 | 90.0(+7.7) | 80.9 | 96.7(+17.1) | 109.9 | 91.7(+2.3) | 85.7 |
| CURL | 99.9(+22.1) | 84.9 | 112.4(+30.1) | 65.9 | 105.1(+25.5) | 78.1 | 108.2(+18.8) | 76.8 |
| LGM | 99.2(+21.4) | 86.5 | 114.6(+32.3) | 67.9 | 104.9(+25.3) | 82.0 | 108.6(+19.2) | 79.2 |

## 4.3 EVALUATION ON DISCRIMINATIVE DGMS COMPRESSION

Finally, we apply the proposed method to compress a discriminative 2-layer DGM with the same MLP parameterization as Alemi et al. (2016). The training loss in Alemi et al. (2016) is denoted by $\mathcal{L}_{VIB}$. We compare the performance of different models using our method (our-VIB) in Eq. (10) and

the other five baselines: (i) combining $\mathcal{L}_{VIB}$ with local distillation loss (Local-VIB), (ii) combining $\mathcal{L}_{VIB}$ with Monte Carlo based marginalized distillation loss (MC-VIB), (iii) model trained with $\mathcal{L}_{VIB}$ only (VIB), (iv) model trained with vanilla cross entropy loss only (Vanilla), and (v) model trained using vanilla cross entropy loss with dropout (Dropout). The experimental results are illustrated in Table 2. We can see that our method beats the baselines in terms of classification accuracy and negative log likelihood (NLL) on Fashion MNIST and SVHN datasets. Besides, our method has lower ECE than all the baselines except for Dropout method.

Table 2: Classification accuracy of different methods averaged over 5 random seeds.

| Methods | #Parameters | Fashion MNIST | | | SVHN | | |
|---------|-------------|----------|-----|-----|----------|-----|-----|
| | | Accuracy | NLL | ECE | Accuracy | NLL | ECE |
| Teacher | 2.60M | 90.40 | 0.4657 | 0.0592 | 83.97 | 0.6589 | 0.0484 |
| Vanilla | 0.40M (6.5×) | 89.69±0.12 | 0.651±0.005 | 0.077±0.001 | 81.07±1.68 | 0.719±0.036 | 0.0294±0.009 |
| Dropout | 0.40M (6.5×) | 89.69±0.13 | 0.653±0.009 | 0.078±0.001 | 76.46±4.95 | 0.831±0.119 | **0.0229±0.005** |
| VIB | 0.40M (6.5×) | 89.70±0.24 | 0.480±0.0122 | 0.064±0.003 | 83.01±0.36 | 0.645±0.007 | 0.0244±0.002 |
| MC-VIB | 0.40M (6.5×) | 90.35±0.08 | 0.459±0.002 | 0.065±0.001 | 83.77±0.32 | 0.689±0.023 | 0.0494±0.0017 |
| Local-VIB | 0.40M (6.5×) | 90.38±0.06 | 0.467±0.001 | 0.059±0.0001 | 83.94±0.01 | 0.656±0.001 | 0.047±0.0003 |
| Our-VIB | 0.40M (6.5×) | **90.43±0.12** | **0.429±0.006** | **0.0576±0.001** | **84.04±0.09** | **0.6307±0.008** | 0.0346±0.001 |

## 5 RELATED WORK

In the past few years, there is a line of work on knowledge distillation (KD) for different DGMs. Most of existing approaches focus on vanilla DNNs or multi-target DNNs Hinton et al. (2015); Chen et al. (2017); Müller et al. (2019); Romero et al. (2014); Zagoruyko & Komodakis (2016). For instance, Chen et al. Chen et al. (2017) proposed a weighted cross entropy loss for multi-class object detection models using KD. Zagoruyko et al. Müller et al. (2019) developed an attention based mechanism to transfer knowledge from the teacher CNN to the student for image recognition tasks. These models can be viewed as DGMs with only one stochastic layer, consisting of one input variable and one or multiple conditionally independent target variables.

Some researchers also study KD for fully visible auto-regressive networks Kim & Rush (2016); Jiao et al. (2019); Oord et al. (2018); Huang et al. (2018). Among them, sequence-level knowledge distillation (SeqKD) Kim & Rush (2016) is a promising strategy that supervises student with the teacher's sequence distribution over the space of all possible sequences. For example, some researchers Lin et al. (2020a) adopted imitation-based knowledge distillation (ImitKD) method to compress autoregressive models for language generation tasks. Tan et al. (2019) proposed a multilingual distillation framework for multilingual Neural Machine Translation (NMT). These auto-regressive models are specific DGMs with multiple stochastic layers, but with no latent variables.

In addition, recent studies apply KD to compress deep generative models with one latent variable. To reduce the number of parameters used in GANs, researchers Aguinaldo et al. (2019); Li et al. (2020) devised new knowledge distillation methods for compressing GANs. Besides, Lee et al. (2020) distilled the learned representation from VAE models to GAN for high-fidelity synthesis. Roy et al. (2018) leveraged VQ-VAE with KD to develop a non-autoregressive machine translation model. However, these works did not apply reparameterization trick to deal with the latent variables.

In summary, the above existing KD methods are mainly focused on specific DGMs, but fail to generalize to the general deep DGMs, especially to those with multiple layers of random variables or complex dependence structures. Different from prior work, we developed a novel unified KD framework for a general deep DGM using reparameterization trick.

## 6 CONCLUSION

This paper proposed a new unified KD framework for deep directed graphical models (DGMs). Specifically, we first adopted reparametrization trick to convert latent variables into deterministic variables with auxiliary variables, resulting in a compact *semi-auxiliary form* of DGM. Then a novel objective that combines the surrogate distillation loss and latent distillation loss was proposed to improve the performance of KD. We further illustrated that our framework is a proper generalization of some existing KD methods. We evaluated the performance of our method on different tasks. The results showed that it can better compress VAE models in a data-free manner than the baselines. It can also achieve high accuracy for discriminative DGMs compression and DGMs with discrete variables.

REPRODUCIBILITY STATEMENT

To replicate our experimental results in Section 4, we have uploaded the source code on GitHub (`https://github.com/papersourcecode/ICLR2022-SourceCode.git`). In our experiments, we adopt seven benchmark datasets: Old Faithful Geyser, IAM online handwriting, SVHN, CIFAR10, CelebA, MNIST, and Fashion-MNIST. The detailed data splitting for training and testing is described in Appendix E. In addition, we present the detailed model configurations and hyperparameter settings in Appendix F. For Proposition 3.1, we provide the detailed proof in Appendix B.

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

## A    Two Naive Methods

We present two naive methods for distilling knowledge from a teacher to a student for a general DGM as follows.

### A.1    Marginalized Distillation

To minimize the desired distillation loss in Eq. (2), one candidate solution is to marginalize over all latent variables, which, however, is generally intractable in DGMs with latent variables. This is because in general there is no analytical solution for the likelihood $p(\boldsymbol{y}|\boldsymbol{x}) = \int p(\boldsymbol{y}, \boldsymbol{z}|\boldsymbol{x})\mathrm{d}\boldsymbol{z}$, much less for $d\left(p_\phi(\boldsymbol{y}|\boldsymbol{x}), p_\theta(\boldsymbol{y}|\boldsymbol{x})\right)$. Thus, we try to use Monte Carlo method to estimate the conditional distribution $p(\boldsymbol{y}|\boldsymbol{x})$ as below

$$p(\boldsymbol{y}|\boldsymbol{x}) = \frac{1}{N}\sum_{n=1}^{N} p(\boldsymbol{y}|\boldsymbol{z}^{(n)}, \boldsymbol{x}), \tag{11}$$

where $\{\boldsymbol{z}^{(n)}\}_{n=1}^{N}$ is a set of samples drawn from $p(\boldsymbol{z}|\boldsymbol{x})$. However, there are two main issues of this method. First, when $\boldsymbol{y}$ consists of more than one random variables (e.g., in deep generative modeling) or it is a continuous variable (e.g, auto-regressive models), $p(\boldsymbol{y}|\boldsymbol{x})$ is often not a simple distribution. As a result, Eq. (2) generally does not have a closed-form solution. Besides, it is not often scalable to the scenario where $\boldsymbol{z}$ contains many latent variables or each latent variable is a high-dimensional vector, since the variance of Monte Carlo estimation may be increased.

The two limitations discussed above suggest that marginalized distillation is only applicable to single-label classification tasks, where latent variables in the deep DGMs must be very low dimensional.

### A.2    Local Distillation

Another naive method is called local distillation, which applies distillation in a layer-wise manner in deep DGMs, as shown in Fig. 3(d). In local distribution, each of student's local component is encouraged to mimic its corresponding teacher's local component in an independent manner. With this method, we can factorize the joint distribution of a deep DGM into a product of a series of conditional distributions, as shown in Eq. (1), so that the distillation loss of each conditional distribution can be calculated separately. Formally, the distillation loss for target variable $\boldsymbol{y}_j$ is given by

$$\mathcal{L}_{y,j} = \mathbb{E}_{p_{data}(\boldsymbol{x})p_\phi(Pa(\boldsymbol{y}_j)|\boldsymbol{x})}\left[d\left(p_\phi(\boldsymbol{y}_j|Pa(\boldsymbol{y}_j), \boldsymbol{x}), p_\theta(\boldsymbol{y}_j|Pa(\boldsymbol{y}_j), \boldsymbol{x})\right)\right] \tag{12}$$

Similarly, the distillation loss for latent variable $\boldsymbol{z}_i$ is

$$\mathcal{L}_{z,i} = \mathbb{E}_{p_{data}(\boldsymbol{x})p_\phi(Pa(\boldsymbol{z}_i)|\boldsymbol{x})}\left[d\left(p_\phi(\boldsymbol{z}_i|Pa(\boldsymbol{z}_i), \boldsymbol{x}), p_\theta(\boldsymbol{z}_i|Pa(\boldsymbol{z}_i), \boldsymbol{x})\right)\right] \tag{13}$$

For the above equations, all the expectations can be estimated using Monte Carlo method, and Monte Carlo samples can be drawn using ancestral sampling from the teacher model.

Finally, we sum up all the local distillation losses and then jointly minimize the overall distillation loss as follows.

$$\mathcal{L}_{kd} = \sum_j \mathcal{L}_{y,j} + \sum_i \mathcal{L}_{z,i} \tag{14}$$

Let $d\left(\cdot, \cdot\right)$ be KL divergence, then Eq. (14) can be simplified as

$$\mathcal{L}_{kd} = \mathbb{E}_{p_{data}(\boldsymbol{x})}\left[KL(p_\phi(\boldsymbol{y}, \boldsymbol{z}|\boldsymbol{x}) \,\|\, p_\theta(\boldsymbol{y}, \boldsymbol{z}|\boldsymbol{x}))\right] \tag{15}$$

The above equation is the KL divergence between the joint probability on all the latent variables and target variables of the teacher and that of the student model. It can be easily proved by factorizing $p(\boldsymbol{y}, \boldsymbol{z}|\boldsymbol{x})$.

For the above Eq. (15), if there are no latent variables in a DGM, and if we choose $d\left(\cdot, \cdot\right)$ to be KL divergence, the local distillation loss can be simply reduced to the distillation loss in Eq. (2) or can be expanded to Eq. (3). However, when there exist latent variables, local distillation will add extra constraints on latent variables, which is redundant and dampens the distillation performance in practice. This is because we are only interested in minimizing the dissimilarity between target variables rather than latent variables.

Next, we provide an insight of why local distillation does not work well in deep DGMs. Take the DGM in Fig. 3(d) as an illustrative example. After distillation, each conditional distribution in the student model may still slightly deviate from the teacher model because of the inferior capacity of the student model or the randomness of optimization process. We call this deviation from student to teacher *imitation error*. Imitation errors will accumulate through multiple layers. For instance, in the imitation process of variable $y_1$, its local distillation loss in Eq. (12) is $\mathcal{L}_{y,1} = \mathbb{E}_{p_{data}(x)p_\phi(z_1|x)}[d(p_\phi(y_1|z_1), p_\theta(y_1|z_1))]$. Even if this local distillation loss for $y_1$ is very well optimized, when the student's latent distribution $p_\theta(z_1|x)$ deviates far from the teacher's distribution, $p_\phi(z_1|x)$, there will be no guarantee that $p_\theta(y_1|x)$ is a good approximation for $p_\phi(y_1|x)$ because $p(y_1|x) = \mathbb{E}_{p(z_1|x)}p(y_1|z_1)$. $p_\theta(y_1|z_1)$ is only well constrained in the domain of $p_\phi(z_1|x)$. As a result, imitation errors may accumulate through layers of latent variables in a deep DGM.

We also conduct a toy experiment to show the error accumulation issue in local distillation. In our experiment, we let student mimic the output distribution of the teacher with $L$-layers latent variables for $L \in 1, \ldots, 20$. Each layer of the teacher follows the Gaussian distribution $p(z_{i+1}|z_i) = \mathcal{N}(\mu(z_i), 0.01I)$, where $\mu(z_i)$ is $z_i^{1.1}$ for $z_i \geq 0$ and $-(-z_i)^{1.1}$ for $z_i < 0$. $p(z_1)$ is a uniform distribution $U[-1, 1]$. The student is parameterized by neural networks with proper residual structure He et al. (2016). Then *density ratio estimation* Nguyen et al. (2010); Sugiyama et al. (2012) is used to measure the KL divergence between the teacher and student. We can observe from Fig. 2 that the accumulated error (KL divergence) grows linearly w.r.t the number of layers for local distillation because each layer of the student deviates from the teacher to an extent. In contrast, the accumulated error (KL divergence) of our method increases slowly.

## B  ABOUT UPPER BOUND OF VANILLA DISTILLATION LOSS

We present the detailed proof of Proposition 3.1 below.

*Proof.* Since the distributions of the auxiliary variables for both student and teacher are the same, i.e., $p_\phi(\epsilon) = p_\theta(\epsilon)$, the surrogate distillation loss as in Eq. (4) can be rewritten as follows.

$$\mathcal{L}_{sd} = \mathbb{E}_{p_\phi(\epsilon)p_{data}(x)p_\phi(y|\epsilon,x)}\left[\log \frac{p_\phi(y|\epsilon,x)}{p_\theta(y|\epsilon,x)}\right]$$
$$= \mathbb{E}_{p_\phi(\epsilon)p_{data}(x)p_\phi(y|\epsilon,x)}\left[\log \frac{p_\phi(y|\epsilon,x)p_\phi(\epsilon)}{p_\theta(y|\epsilon,x)p_\theta(\epsilon)}\right]$$

Notice that $p_\phi(y|\epsilon,x)p_\phi(\epsilon) = p_\phi(y,\epsilon|x) = p_\phi(\epsilon|y,x)p_\phi(y|x)$. Similarly, it also holds that $p_\theta(y|\epsilon,x)p_\theta(\epsilon) = p_\theta(\epsilon|y,x)p_\theta(y|x)$. Thus, we can further rewrite the surrogate distillation loss $\mathcal{L}_{sd}$ as below.

$$\mathcal{L}_{sd} = \mathbb{E}_{p_{data}(x)p_\phi(\epsilon|y,x)p_\phi(y|x)}\left[\log \frac{p_\phi(\epsilon|y,x)}{p_\theta(\epsilon|y,x)} + \log \frac{p_\phi(y|x)}{p_\theta(y|x)}\right]$$
$$= \mathbb{E}_{p_{data}(x)p_\phi(y|x)}\left[KL\left(p_\phi(\epsilon|y,x) \parallel p_\theta(\epsilon|y,x)\right)\right] + \mathbb{E}_{p_{data}(x)}\left[KL\left(\log p_\phi(y|x) \parallel p_\theta(y|x)\right)\right]$$

Since the first term is also non-negative, it further holds that

$$\mathcal{L}_{sd} \geq \mathbb{E}_{p_{data}(x)}\left[KL\left(\log p_\phi(y|x) \parallel p_\theta(y|x)\right)\right] = \mathcal{L}_{kd}$$

which finishes our proof.

$\square$

## C  ALGORITHM SUMMARY

We summarize our knowledge distillation method for a general DGM in Algorithm 1. Please note that we use $v_i$ to denote a non-root random variable in DGMs, which is either a latent variable $z_i$ or

a target variable $\boldsymbol{y}_i$. $\boldsymbol{v}_{\phi,i}$ and $\boldsymbol{v}_{\theta,i}$ denote a sampling point of node $\boldsymbol{v}_i$ in the teacher DGM and the student, respectively. It can be observed from the distillation process that we only need to traverse the graph once to compute both the surrogate distillation loss and the latent distillation losses. Line 9 is our latent distillation loss. In Line 10-12, we describe the latent variables of the teacher and student model that are shared indirectly via sharing the auxiliary variables $\boldsymbol{\epsilon}$. Line 14 computes the surrogate distillation loss. Line 15-16 describe that the target variables of teacher and student model are shared directly. Line 19 presents the back propagation and gradient descent.

---

**Algorithm 1** Our Distillation Method

---

**Require:** teacher DGM $\mathcal{G}_\phi$, student DGM $\mathcal{G}_\theta$, learning rate $\eta$, hyperparameter $\lambda$, empirical distribution of input dataset $p_{data}(\boldsymbol{x})$ (only required when $\boldsymbol{x} \neq \emptyset$)
**Ensure:** distilled student DGM $\mathcal{G}_\theta$
 1: **while** $\theta$ not converged **do**
 2:     sample mini-batched $\boldsymbol{x}$ from $p_{data}(\boldsymbol{x})$ // (only required when $\boldsymbol{x} \neq \emptyset$)
 3:     set $L_{our} = 0$
 4:     **while** $\mathcal{G}$ not fully traversed **do**
 5:         grab variable $\boldsymbol{v}_i$ where $Pa(\boldsymbol{v}_i)$ have all been sampled
 6:         set $p_{\phi,i} = p_\phi(\boldsymbol{v}_i|Pa(\boldsymbol{v}_i), \boldsymbol{x})$
 7:         set $p_{\theta,i} = p_\theta(\boldsymbol{v}_i|Pa(\boldsymbol{v}_i), \boldsymbol{x})$
 8:         **if** $\boldsymbol{v}_i$ is latent variable **then**
 9:             $\mathcal{L}_{our} \leftarrow \mathcal{L}_{our} + \lambda d(p_{\phi,i}, p_{\theta,i})$
10:             sample $\boldsymbol{\epsilon}_i$ from $p_\phi(\boldsymbol{\epsilon}_i)$
11:             sample $\boldsymbol{v}_{\phi,i}$ from $p_{\phi,i}$ according to $\boldsymbol{\epsilon}_i$ // set $\boldsymbol{v}_{\phi,i} = g_\phi(Pa(\boldsymbol{v}_i), \boldsymbol{x}, \boldsymbol{\epsilon}_i)$
12:             sample $\boldsymbol{v}_{\theta,i}$ from $p_{\theta,i}$ according to $\boldsymbol{\epsilon}_i$ // set $\boldsymbol{v}_{\theta,i} = g_\theta(Pa(\boldsymbol{v}_i), \boldsymbol{x}, \boldsymbol{\epsilon}_i)$
13:         **else** $\{\boldsymbol{v}_i$ is target variable$\}$
14:             $\mathcal{L}_{our} \leftarrow \mathcal{L}_{our} + d(p_{\phi,i}, p_{\theta,i})$
15:             sample $\boldsymbol{v}_{\phi,i}$ from $p_{\phi,i}$
16:             $\boldsymbol{v}_{\theta,i} \leftarrow \boldsymbol{v}_{\phi,i}$
17:         **end if**
18:     **end while**
19:     $\theta \leftarrow \theta - \eta \frac{\partial \mathcal{L}_{our}}{\partial \theta}$
20: **end while**

---

## D   ILLUSTRATION OF GRAPHICAL MODELS FOR DIFFERENT APPLICATIONS

Fig. 6 shows different forms of deep DGMs used in our experiments.

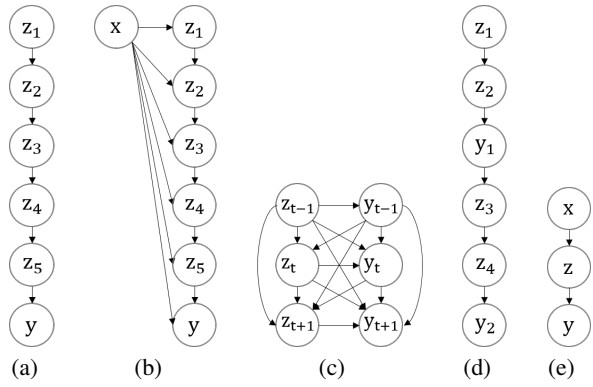

(a)        (b)        (c)        (d)    (e)

Figure 6: DGMs used in our experiments. (a) 5-layer DGM without input variables. (b) 6-layer DGM with one binary input variable indicating which domain the sample comes from. (c) DGM with as many latent variables and target variables as the sequence length and no input variable. (d) 5-layer DGM with two target variables and no input variable. (e) 2-layer DGM with one input variable
.

## E  DATASETS

To evaluate the performance of the proposed distillation method, we conduct experiments on the following benchmark datasets.

- **Old Faithful Geyser** Härdle et al. (1991): this is 2-dimensional Geyser data. We use all the 272 samples for training.
- **IAM Online Handwriting** Liwicki & Bunke (2005): it has 13040 samples and each sample is a sequence of $(x, y)$ coordinate together with binary indicator of pen up/down. We use all the examples for training.
- **SVHN** Netzer et al. (2011): This dataset is about house numbers in street views with 1 categorical label of 10 classes. We use 73257 images for training and 26032 images for testing respectively.
- **CIFAR10** Krizhevsky et al. (2009): This dataset consists of 60000 natural world images, with 1 categorical label of 10 classes. The data is split into 50000 and 10000 images for training and testing.
- **CelebA** Liu et al. (2015): It contains $182, 732$ RGB $128 \times 128 \times 3$ images of celebrity faces. We use 162770 images for training and 19962 images for testing.
- **Fashion-MNIST** Sohn et al. (2015): This dataset is about Grey-scale dressing images. We use 60000 images for training and 10000 images for testing in our experiment.

## F  MODEL CONFIGURATIONS AND HYPERPARAMETER SETTINGS

We summarize the detailed model configurations and hyperparameter settings for the proposed method in different tasks below.

For all the tasks, we choose $d(\cdot, \cdot)$ in the surrogate distillation loss $\mathcal{L}_{sd}$ to be KL divergence. When latent variables are discrete, $d(\cdot, \cdot)$ in the latent distillation loss $\mathcal{L}_{z,i}$ is chosen to be KL divergence and $\lambda$ is set to 1. When latent variables are continuous, we choose $d(\cdot, \cdot)$ in $\mathcal{L}_{z,i}$ to be squared 2-Wasserstein distance because it has analytical solution on two Gaussian distributions. In addition, we normalize $\mathcal{L}_{z,i}$ w.r.t. the size of vector $z_i$. We empirically found that normalized latent distillation loss works slightly better than unnormalized one. In all the following experiments, $\lambda$ is chosen after a simple grid search.

The main training configurations for different tasks are summarized in the Table 3 below.

Table 3: Configurations summary

| Configuration | VAE Compression | VRNN Compression | HM Compression | Discriminative DGM Compression |
|---|---|---|---|---|
| Teacher size | 5.4M | 12M | 354 | 2.6M |
| Student size | 0.01M-1.4M | 2.4M | 102 | 0.40M |
| Optimizer | Adam | Adam | Adam | Adam |
| Learning rate | 1e-4 | 1e-4 | 1e-2 | 1e-3 |
| Weight decay | 1e-4 | 1e-4 | 1e-4 | 1e-4 |
| Batch size | 128-256 | 32 | 272 | 256 |

### F.1  DATA-FREE VAE COMPRESSION

In our experiments, we distill a VAE with the same DGM and parameterization as the Ladder VAE Sønderby et al. (2016). The DGM structure of its generative model (decoder) is shown in Fig. 6(a). Slightly different from Sønderby et al. (2016), we use Convolutional Neural Network (CNN) to parameterize both the bottom-up and top-down network in VAE, instead of simple MLP. In addition, we adopt spatial latent variables (latent variables are spatially arranged, as the feature map in CNN) in the neural networks. Each CNN is composed of consecutive residual blocks He et al. (2016). We fix the size of the teacher model to 5.4M parameters, with almost symmetric structure on the bottom-up network and top-down network. To demonstrate the robustness of our method, we vary the student model size with different compression times.

In the experiments, we set the depth of latent variables to 32 for both the teacher and student VAE. The hidden dimension of residual blocks varies from 4 to 64 in the student model, and is set to 128 for the teacher VAE. Mini-batch sizes are set to 256 for SVHN and CIFAR10 , and 128 for CelebA. When VAE is trained from scratch, we adopt linear $\beta$ annealing, which is called *warm up* in Sønderby et al. (2016) for first half of the iterations to stabilize model learning. For CelebA dataset, we resize the images to the shape $64 \times 64 \times 3$. Model training is stopped after 57k, 46k, 63k iterations for SVHN, CIFAR10, and CelebA, respectively.

### F.2    VAE Continual Learning

In our experiment, we still use the same 5-layer VAE as the above VAE compression experiment. Following Rao et al. (2019), we add an additional discrete input variable to the graph for the sake of indicating where the data comes from. As a result, our VAE model is turned to a 6-layer Conditional VAE, as illustrated in Fig. 6(d).

For VAE continual learning, we have the following distillation loss.

$$
\begin{aligned}
\tilde{L}_{continual} &= \mathbb{E}_{\frac{1}{2}(p_{old}(\boldsymbol{y})+p_{new}(\boldsymbol{y}))}\left[-\log p_\theta(\boldsymbol{y})\right] \\
&= \frac{1}{2}(\mathbb{E}_{p_{old}(\boldsymbol{y})}\left[-\log p_\theta(\boldsymbol{y})\right] + \mathbb{E}_{p_{new}(\boldsymbol{y})}\left[-\log p_\theta(\boldsymbol{y}))\right] \\
&= \frac{1}{2}(KL[p_{old}(\boldsymbol{y}) \parallel p_\theta(\boldsymbol{y})] + KL[p_{new}(\boldsymbol{y}) \parallel p_\theta(\boldsymbol{y})]) + c
\end{aligned}
\tag{16}
$$

where $c$ is a constant independent of $\theta$. $p_{old}(\boldsymbol{y})$ and $p_{new}(\boldsymbol{y})$ are the empirical distribution of old and new dataset, respectively. For the above loss, since $p_{old}(\boldsymbol{y})$ is not accessible, we use the old learned VAE's distribution $p'_\theta(\boldsymbol{y})$ as a replacement. Accordingly, we replace $KL[p_{old}(\boldsymbol{y}) \parallel p_\theta(\boldsymbol{y})]$ with $KL[p'_\theta(\boldsymbol{y}) \parallel p_\theta(\boldsymbol{y})]$. This results in our desired distillation loss, transferring knowledge from teacher $p'_\theta(\boldsymbol{y})$ to student $p_\theta(\boldsymbol{y})$, which, however, is intractable as discussed earlier. To tackle this issue, we use our distillation loss $\mathcal{L}_{our}$ instead. As a result, the total loss for continual learning is rewritten as

$$
L_{continual} = \frac{1}{2}(L_{our} + KL[p_{new}(\boldsymbol{y}) \parallel p_\theta(\boldsymbol{y})])
\tag{17}
$$

In this experiment, we follow the model configurations as the above VAE compression, except that we resize images to $32 \times 32 \times 3$ and do not use running statistics for batch normalization layers during evaluation.

### F.3    Data-free VRNN Compression

In this experiment, we build and distill a VRNN with the same DGM and parameterization as in Fraccaro et al. (2016), as illustrated in Fig 6(c). VRNN consists of as many latent variables and target variable as its sequence length. We fix the number of hidden layer in all densely connected networks to 1, and fix the RNN to be 1-layer LSTM as in Fraccaro et al. (2016). The dimension of latent variables $\boldsymbol{z}$ is set to 16. For the teacher model, the hidden dimension of LSTM is set to 1200. Feature size of target variables, latent variables, and hidden dimension of all densely connect networks are set to 512. For the student model, the hidden dimension of LSTM is set to 600. Feature dimension of target/latent variables, and hidden dimension of all densely connect networks are set to 128.

### F.4    Data-free HM Compression

Helmholtz Machine (HM) Neal (1992); Saul et al. (1996) is a classical generative modeling technologyGregor et al. (2014); Bornschein & Bengio (2014); Vértes & Sahani (2018); Bornschein et al. (2016) that consists of: 1) Sigmoid Belief Network (SBN) as generative graphical model; 2) another SBN as amortized inference network; 3) Wake-Sleep learning strategy. The classical HM has only one target variable and each layer is parameterized by a linear transformation. In our experiment, in order to demonstrate the applicability of our distillation method, we extend it to a 5-layer DGM with two target variables, as shown in Fig. 6(d). The inference network is also modified to match the true posterior dependency structure. Following the similar idea in Krishnan et al. (2015) and Bornschein & Bengio (2014), each conditional distribution in both generative and inference network are parameterized by 2-layer MLP instead of linear transformation, which turns our HM to *deep HM*.

In our experiment, we fix latent variables to two-dimensional vectors with binary elements. The hidden dimension of MLP is set to 8 and 2 for teacher and student model respectively.

### F.5 DISCRIMINATIVE DEEP DGM COMPRESSION

In this experiment, we distill a Discriminative Deep DGM with the same 3-layer MLP parameterization as Alemi et al. (2016). We fix the hyperparameter $\beta$ in original training loss to be 1e-3. Latent variable is set to be 256 dimensional vector. The hidden size of MLP is set to 1024 and 256 for the teacher and the student model, respectively. Models in our experiment are trained for $46k$ iterations for MNIST, Fashion-MNIST dataset, and are trained for $57k$ iterations for SVHN dataset. Images in SVHN dataset are first turned to grayscale.

## G EXPERIMENTS ON VAE COMPRESSION

### G.1 SIMILARITY BETWEEN TEACHER AND STUDENT

Fig. 7 compares the FID between samples generated from the student and teacher for different methods. We can observe that our method beats the baselines: local distillation and trained from scratch.

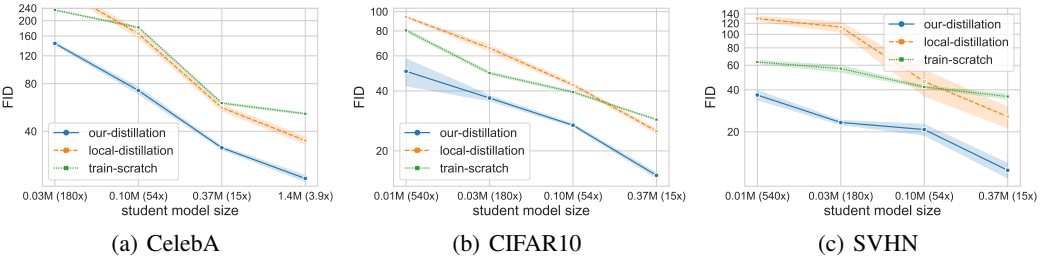

|       (a) CelebA        |       (b) CIFAR10        |        (c) SVHN        |

Figure 7: FID between samples generated by the student and teacher averaged over 3 random seeds.

### G.2 QUALITATIVE RESULTS ON CELEBA

We also conduct experiments to compare the generated samples by different knowledge distillation (KD) methods on the CelebA dataset, as illustrated in Fig. 8. It can be observed from Fig. 8(c) that images generated by our method are high-fidelity, as good as ground truth and those generated by the teacher, which means our method can imitate teacher's performance very well. However, a limited-size VAE model is incapable of learning such a complicated distribution from scratch on its own. Hence, the images generated by the student VAE trained from scratch are low quality, as illustrated in Fig. 8(d).

### G.3 HYPERPARAMETER STABILITY IN THE LOSS

We further demonstrate the hyper-parameter stability of our method. Fig. 9 shows the relationship between $\lambda$ and FID score. It can be observed that as $\lambda$ varies from to 0.0 to $1e3$, the FID score of our distillation method does not vary too much. It means that our method is stable and robust to the hyper-parameter $\lambda$.

## H EVALUATION ON DATA-FREE HELMHOLTZ MACHINE COMPRESSION

We apply the proposed approach to compress a 5-layer HM trained with Wake-Sleep algorithm to a smaller HM in a data-free manner. We still compare it with the baselines: training from scratch and local distillation. When a model is trained from scratch instead of distillation, we add an inference model (encoder) with symmetric structure and almost the same number of parameters as its generative model. The experimental results are shown in Fig. 10.

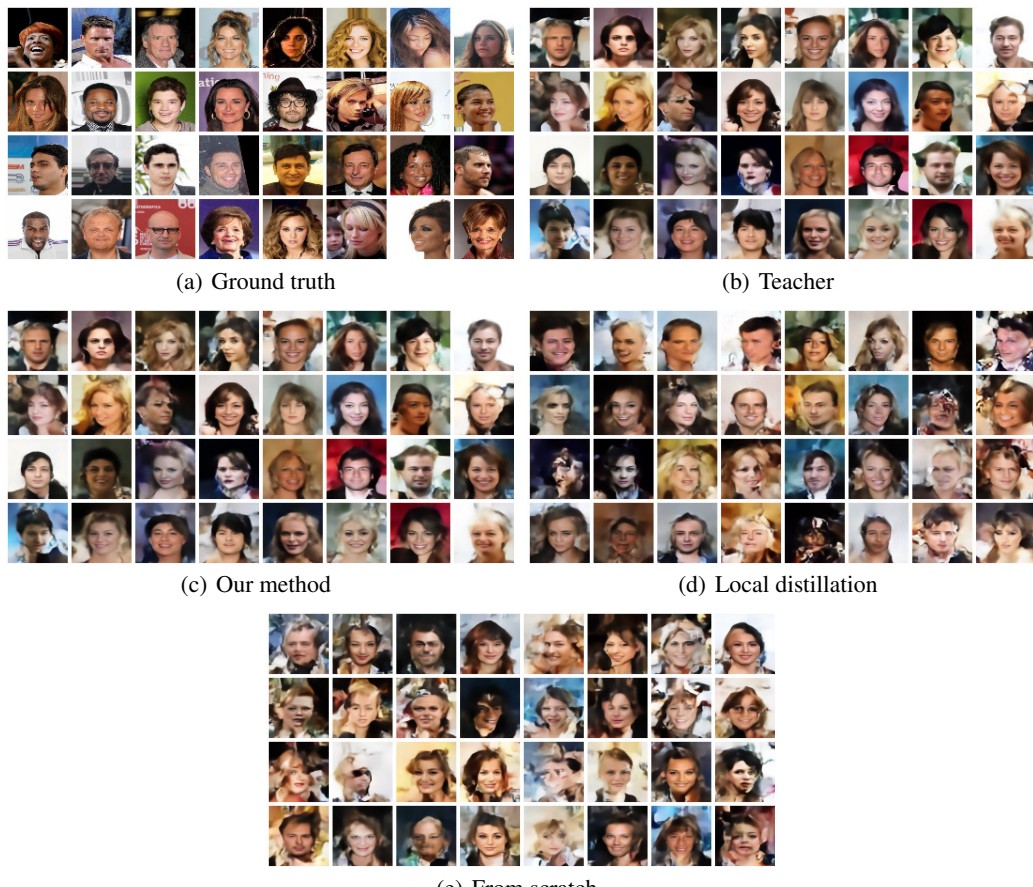

(a) Ground truth

(b) Teacher

(c) Our method

(d) Local distillation

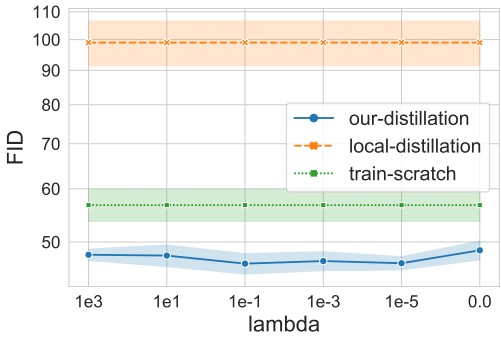

(e) From scratch

Figure 8: Images generated by VAE trained on CelebA dataset. (a) Ground truth dataset. (b) Teacher VAE (5.4M parameters) trained from scratch. (c) Student VAE (1.4M parameters, 3.9 times smaller) distilled by our method. (d) Student VAE distilled using local distillation. (e) Student VAE trained from scratch.

Figure 9: Effect of $\lambda$ in Eq. (7) on the performance of our method.

We can observe that our distillation method still outperforms the student model trained from scratch, just as VAE and VRNN compression. In addition, it shows that optimizing a capacity-limited HM may not learn a powerful model that can generate high-fidelity samples. In contrast, our method enables a small student HM to attain good performance as a larger teacher model.

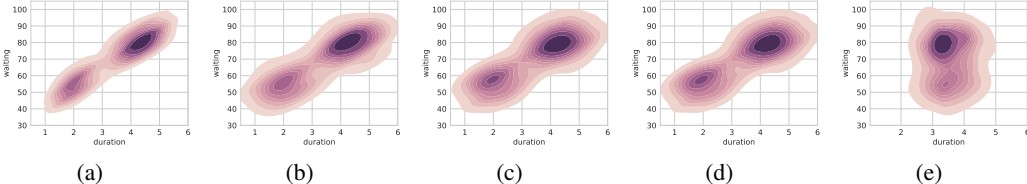

|     |     |     |     |     |
| --- | --- | --- | --- | --- |
| (a) | (b) | (c) | (d) | (e) |

Figure 10: Generation distribution of HMs. We performed and plot kernel density estimation using 272 generated samples for each HM. **(a)** Ground truth. **(b)** Teacher HM (354 parameters) trained from scratch. **(c)** Student HM (102 parameters, 3.5 times smaller) distilled using our method. **(d)** Student HM distilled using local distillation. **(e)** Student HM trained from scratch.

In our experimental results, we find that there is tiny difference between our method and local distillation, partly because this 2-dimensional modeling task is too easy for highly flexible HMs to learn. Both our method and local distillation can converge after a few iterations. Hence, they have comparable performance to each other.

# I    SCALABILITY ON LARGE NUMBER OF DISCRETE VARIABLES

Our proposed method can also be scalable to a DGM with a large number of discrete latent variables. We take a DGM with $L$ (e.g, 20) layers of discrete latent variables as an example to demonstrate the good performance of our approach, as shown in Fig. 11. In our experiment, we let student mimic the output distribution of the teacher with $L$-layers latent variables for $L \in 1, \ldots, 20$. Each layer of the teacher follows the binomial distribution $p(\boldsymbol{z}_{i+1}|\boldsymbol{z}_i) = \mathcal{B}(1000, \boldsymbol{p}(\boldsymbol{z}_i))$, where $\boldsymbol{p}(\boldsymbol{z}_i) = (\frac{\boldsymbol{z}_i}{1000})^{1.2}$; $p(\boldsymbol{z}_1)$ is a uniform distribution $U[0, 1]$. The student is parameterized by neural networks with a residual structure He et al. (2016). Then *density ratio estimation* Nguyen et al. (2010); Sugiyama et al. (2012) is used to measure the KL divergence between the teacher and the student. We can observe from Fig. 11 that even if all the latent variables are discrete, our method still has the capability of mitigating the imitation error occurred in the shallow layers and achieves a very good imitation performance. For the local distillation method, however, its imitation errors are accumulated as the number of layers increases.

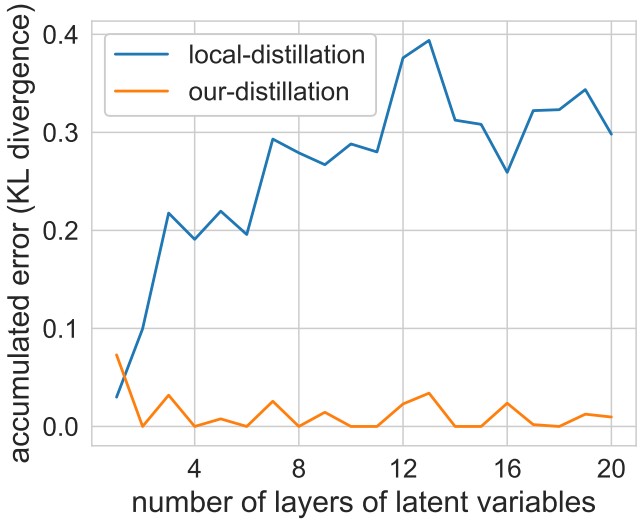

Figure 11: A toy example of accumulated error (KL divergence) between the teacher and student for local distillation and our method in a DMG with discrete random variables.

## J    DISCUSSION AND FUTURE WORK

We further discuss the limitation of the proposed framework. Our framework is only applied to DGMs where latent variables can be reparameterized. However, we would like to point out that the definition of reparameterization trick in our framework is slightly different from the classical one for VAE training Kingma & Welling (2013). Our reparameterization is more general than that for propagating gradients for model training. Specifically, reparameterization in our work is used for compacting the DGMs to semi-auxiliary forms rather than propagating gradients. Different from the classical reparameterization for model training that requires continuous latent variables, ours can be applied to a wider range of variables $z$, including both continuous and discrete variables. In addition, the transformation function $g(\cdot)$ in our framework can be either differentiable or non-differentiable. Hence, our method can be applied to much more DGMs than the classical one. Table 4 compares the different requirements for our method and the classical reparameterization trick.

Table 4: Different requirements for our method and the classical reparameterization trick.

| Requirements | Traditional reparameterization | Our framework |
|---|---|---|
| Transformation function $g(\cdot)$ | differentiable | differentiable / non-differentiable |
| Latent variable $z$ | continuous | continuous / discrete |

Another limitation of our framework is that it needs to consist of the same number of random variables (these variables can be arranged in different structures). For future work, we plan to extend the proposed method to deal with more difficult cases where the teacher and students have different number of random variables

