# OpenReview forum: "A Unified Knowledge Distillation Framework for Deep Directed Graphical Models"
_ICLR.cc/2022/Conference — ICLR 2022 Submitted_

### Official Review · Reviewer_1kU6 · 2021-10-30

**Correctness:** 3
**Technical Novelty And Significance:** 2
**Empirical Novelty And Significance:** 2
**Recommendation:** 6
**Confidence:** 3

**Main Review:**

Pros:
1. The authors do a good job of giving basic preliminaries and of ruling out the naïve marginalized distillation and local distillation approaches
2. After the compact DGM reductions for both the teacher and student networks, each target variable has direct dependence on the input x and prior y_i’s. This is a neat approach.
3. Proposition 3.1 looks correct to me, when KL divergence is chosen.

Concerns & questions:
1. Just verifying: The reduction to a compact semi-auxiliary form is a novel contribution of this work, correct? (Fig. 1c, 1d, 3e)
2. How do we get the correct choice of deterministic transformation g(.) for the reparameterization trick of the original teacher network? p_\phi() is a neural network, so I am curious how to get g(.) without loss of accuracy. (Algo.1, lines 11-12). Please give a detailed example.
3. The chain error accumulation is reduced as the number of layers reduces in the semi-auxiliary form, right (pg5, para2) ? Or is there any other rationale to it?
4. I wonder how the performance is with only using L_{sd} in eq(7), \lambda=0 setting?
5. I am a bit confused about the VAE compression expts. Each layer is considered a latent variable `z’, I presume. In this case, what is the student network chosen (I might have missed it)? Also, curious to know, how well the proposed L_{sd} loss works with \lambda=0 setting. Kindly clarify this expt.


**Summary Of The Paper:**

The authors propose an unified Knowledge Distillation technique for general deep directed graphical models. They use the reparameterization trick on the intermediate latent variables of the original DGM network and the student network. This converts the networks to a compact semi-auxiliary form. Then they use a surrogate distillation loss (combined with latent loss) to reduce the error accumulations over the chain of random variables. They discuss the similarity of their technique with others and demonstrate its performance for 3 applications.

**Summary Of The Review:**

The work is good and the paper is well written. I feel the contribution of the work is not that novel. My confidence in evaluation will increase once the authors address the concerns raised above.

---

> ### Author Response · Authors · 2021-11-21
> **Response to Reviewer 4 (Part 1/3)**
>
> **Q4.1**: On the reduction to a compact semi-auxiliary form is a novel contribution of this work
>
> **A4.1:** Yes. This is one of the main contributions in our work. To our best knowledge, this is the first work to develop a new unified knowledge distillation framework for deep DGMs with multiple-layers latent variables using reparameterization trick.
>
> **Q4.2:** On the novelty of this work
>
> **A4.2:** We would like to note that there are some differences between VAE training and our KD using reparameterization trick. The definition of our reparameterization is more general than that for propagating gradients for VAE model training[Kingma 2013]. Specifically, reparameterization in our work is used for compacting the DGMs to semi-auxiliary forms rather than propagating gradients. Different from the classical reparameterization for model training that requires continuous latent variables, ours can be applied to both *continuous and discrete variables*. In addition, the transformation function $g(\cdot)$ in our framework can be either differentiable or non-differentiable. Hence, our method can be applied to much more DGMs than the classical one. Table 7 summarizes the different requirements for our method and the classical reparameterization trick. Besides, we discuss how to apply our method to DGMs with discrete latent variables in our revised manuscript. We further conduct experiments on Helmholtz Machine (HM) model with discrete variables to demonstrate the good performance of our method, as introduced in Section 4.1. We have added the detailed discussions in the appendix in our revised manuscript.
>
> Table 7: Different requirements for our method and the classical reparameterization trick.
>
> | Requirements                       | Traditional reparameterization | Our framework                       |
> |------------------------------------|--------------------------------|-------------------------------------|
> | Transformation function $g(\cdot)$ | differentiable                 | differentiable / non-differentiable |
> | Latent variable $\boldsymbol{z}$   | continuous                     | continuous / discrete               |
> |    |              |            |
>
> Finally, we would like to emphasize the main contributions of this work below.
>
> - To the best of our knowledge, this work is the first to develop a novel unified knowledge distillation (KD) framework for deep DGMs, especially for those with multiple layers of continuous/discrete latent variables.
> - We offer a theoretical proof that the proposed surrogate distillation loss is an upper bound of that of the regular KD.
> - We also present the subtle connections between our method and some existing KD approaches.
> - We conduct extensive experiments on different kinds of DGMs, such as hierarchical VAE, complicated VRNN, and HM with discrete latent variables, to evaluate the performance of the proposed framework. Experimental results on multiple benchmark datasets show that our approach can not only achieve high accuracy for data-free deep DGMs compression but also improve the continual learning performance of VAEs based on knowledge distillation.
>
> [Kingma 2013] Kingma, Diederik P., and Max Welling. "Auto-encoding variational bayes." arXiv preprint arXiv:1312.6114 (2013).

---

> > ### Author Response · Authors · 2021-11-21
> > **Response to Reviewer 4 (Part 2/3)**
> >
> >
> > **Q4.3:** On the correct choice of deterministic transformation $g(.)$ for the reparameterization trick of the original teacher network
> >
> > **A4.3:** According to literature[Kingma 2013 a], the choice of $g(.)$ is straightforward. Suppose $\boldsymbol{z}_i$ is to be reparameterized, the choice of $g(.)$ is only related to the distribution of $p(\boldsymbol{z}_i|\mathit{Pa}(\boldsymbol{z}_i),\boldsymbol{x})$. For instance, if it satisfies a  Gaussian distribution, $p(\boldsymbol{z}_i|\mathit{Pa}(\boldsymbol{z}_i),\boldsymbol{x}) = \mathcal{N}(\mu(\mathit{Pa}(\boldsymbol{z}_i),\boldsymbol{x}), \sigma^2(\mathit{Pa}(\boldsymbol{z}_i),\boldsymbol{x}))$, $g(.)$ can be chosen as $\boldsymbol{z}_i = g(\boldsymbol{\epsilon}, \mathit{Pa}(\boldsymbol{z}_i),\boldsymbol{x}) =\boldsymbol{\epsilon} \times \sigma(\mathit{Pa}(\boldsymbol{z}_i),\boldsymbol{x}) + \mu(\mathit{Pa}(\boldsymbol{z}_i),\boldsymbol{x})  $, where $\boldsymbol{\epsilon}$ is sampled from unit Gaussian distribution. For another case with discrete variables, if it yields to a Bernoulli distribution, $p(\boldsymbol{z}_i|\mathit{Pa}(\boldsymbol{z}_i),\boldsymbol{x})= \mathcal{B}(1,p(\mathit{Pa}(\boldsymbol{z}_i),\boldsymbol{x}))$, $g(.)$ can be chosen as $\boldsymbol{z}_i = g(\boldsymbol{\epsilon}, \mathit{Pa}(\boldsymbol{z}_i),\boldsymbol{x}) = \mathbf{1} _{ p(\mathit{Pa}(\boldsymbol{z}_i),\boldsymbol{x}) > \boldsymbol{\epsilon}}$, where $\boldsymbol{\epsilon}$ is sampled from 0-1 uniform distribution. For more details, please refer to literature[Kingma 2013 a, Kingma 2013 b].
> >
> >
> > [Kingma 2013 a]: Kingma, Diederik P., and Max Welling. "Auto-encoding variational bayes." arXiv preprint arXiv:1312.6114 (2013).
> >
> > [Kingma 2013 b]: Kingma, D. P. (2013). Fast gradient-based inference with continuous latent variable models in auxiliary form. arXiv preprint arXiv:1306.0733.
> >
> > **Q4.4:** On the chain error accumulation is reduced as the number of layers reduces in the semi-auxiliary form. Any other rationale?
> >
> > **A4.4:** Yes. The chain error accumulation is highly correlated with the number of layers, as illustrated in Fig. 2. This is the key motivation of our work that leverages reparameterization trick to convert the latent variables to deterministic variables for making the DGMs shallower.
> >
> > We also provide a detailed explanation of error accumulation in local distillation. Suppose $p_\theta(\boldsymbol{z} _{j-1}|\boldsymbol{x})$ is different from $p_\phi(\boldsymbol{z} _{j-1}|\boldsymbol{x})$, when we try to learn $p_\theta(\boldsymbol{z} _j|\boldsymbol{z} _{j-1},\boldsymbol{x})$ by minimizing local distillation loss $\mathbb{E} _{p _{data}(\boldsymbol{x})p_\phi(\boldsymbol{z} _{j-1}| \boldsymbol{x})} \left[d( p_\phi(\boldsymbol{z} _j| \boldsymbol{z} _{j-1},\boldsymbol{x}) , p_\theta(\boldsymbol{z} _j| \boldsymbol{z} _{ j-1},\boldsymbol{x}) ) \right]$, there will be a mismatch between its training set $p_\phi(\boldsymbol{z} _{j-1}|\boldsymbol{x})$ and testing set $p_\theta(\boldsymbol{z} _{j-1}|\boldsymbol{x})$, leading to the inferior performance of $p_\theta(\boldsymbol{z} _j|\boldsymbol{z} _{j-1},\boldsymbol{x})$. The inferior performance of $p_\theta(\boldsymbol{z}_j|\boldsymbol{z} _{j-1},\boldsymbol{x})$, together with the mismatch between $p_\phi(\boldsymbol{z} _{j-1}|\boldsymbol{x})$ and $p_\theta(\boldsymbol{z} _{j-1}|\boldsymbol{x})$, will further enlarge the mismatch between $p_\theta(\boldsymbol{z} _j|\boldsymbol{x})$ and $p_\phi(\boldsymbol{z} _j|\boldsymbol{x})$. As a result, it causes massive negative impact on the learning of the succeeding random variables $p_\phi(\boldsymbol{z} _{>j}|\boldsymbol{x})$. This vicious circle describes the detailed process of *error accumulation*.

---

> > > ### Author Response · Authors · 2021-11-21
> > > **Response to Reviewer 4 (Part 3/3)**
> > >
> > >
> > > **Q4.5:** On the performance with only using $L_{sd}$ in Eq.(7), $\lambda=0$ setting.
> > >
> > > **A4.5:** We have evaluated the performance of the proposed method when $\lambda=0$, as illustrated in Fig. 9 in the appendix. It can be observed from it that as $\lambda$ varies from to 0.0 to $1e3$, the FID score of our method does not change too much. It means that our method is stable and robust to the hyper-parameter $\lambda$. When $\lambda = 0$, it still performs better than the baselines.
> > >
> > >
> > > **Q4.6:** On confused about the VAE compression expts. Each layer is considered a latent variable $z$, I presume. In this case, what is the student network chosen? Kindly clarify this expt.
> > >
> > > **A4.6:** In our VAE compression experiment, we use a hierarchical VAE with 5 *random* latent variables, as shown in Fig. 6 (a) in the appendix. Specifically, each conditional distribution, such as $p(\boldsymbol{z}_2|\boldsymbol{z}_1)$, is parameterized by a residual deep neural network with multiple deterministic(computation) layers. We then alter the structure of those residual deep neural network for reducing the number of parameters of the student. In our experiments, we only change the width of the residual deep neural network for the sake of simplicity.

---

> > > > ### Comment · Reviewer_1kU6 · 2021-11-25
> > > > **Any experiments on a more complex DGM ?**
> > > >
> > > > It would be great to see results on a DGM apart from VAE. There are many complex graphical models occurring in the medical domain, adapting it to a DGM and then demonstration of this technique will be helpful in evaluation. In complex DGM, I presume that the chain error propagation will dominate and it will be helpful to see the resilience of this technique in those settings.

---

> > > > > ### Author Response · Authors · 2021-11-25
> > > > > **Response to experiments on a more complex DGM**
> > > > >
> > > > > **Q**: On experiments on a more complex DGM
> > > > >
> > > > > **A**: Our method can deal with more complex DGMs than VAE. Fig. 6 (c) in the appendix shows an example of a complicated DGM, called variational recurrent neural networks (VRNN), that contains many latent variables and target variables. We have conducted experiments to evaluate the VRNN model in Section 4.1. Evaluation results on two benchmark datasets demonstrate that the proposed KD method can handle complicated DGMs, and it outperforms the other baselines. For more details, please refer to Section 4.1. For future work, we would like to apply our KD method to the complicated DGMs in the medical domain suggested by the Reviewer.

---

> > > ### Comment · Reviewer_1kU6 · 2021-11-25
> > > **Choice of g(.)**
> > >
> > > I understand the work of Kingma 2013. Perhaps, my question was not clear. What if the underlying distribution is not Gaussian or Bernoulli? Some distribution where we might not be able to separate out the randomness. I am curious on how will this case be handled? Any ideas?

---

> > > > ### Author Response · Authors · 2021-11-25
> > > > **Response to Choice of g(.)**
> > > >
> > > > Thanks a lot for the comments.
> > > >
> > > > **A**: Admittedly, there exist a few distributions that are hard to be reparameterized (randomness not separable). For these cases, we may adopt a proper approximation method, such as gamma distribution in literature [Knowles 2015]. In addition, regarding how $g(\cdot)$ is chosen for a wide range of continuous distributions, please refer to literature [Mohamed 2015]. For discrete distributions, one simple way to reparameterize any distribution with a finite support is to divide $U[0,1]$ into as many pieces as the cardinality of the support. We will discussed the limitation of our work in the final version.
> > > >
> > > > **References**
> > > >
> > > > [Knowles 2015] Knowles D A. Stochastic gradient variational Bayes for gamma approximating distributions[J]. arXiv preprint arXiv:1509.01631, 2015.
> > > >
> > > > [Mohamed 2015] Mohamed, S. “Machine Learning Trick of the Day (4): Reparameterisation Tricks.” The Spectator, Shakir’s Machine Learning Blog. 29 Oct 2015. http://blog.shakirm.com/2015/10/machine-learning-trick-of-the-day-4-reparameterisation-tricks/.

---

### Official Review · Reviewer_twdH · 2021-11-02

**Correctness:** 3
**Technical Novelty And Significance:** 2
**Empirical Novelty And Significance:** 2
**Recommendation:** 5
**Confidence:** 3

**Main Review:**

Strengths: The paper proposes a knowledge distillation framework for deep DGMs on various applications.

Weaknesses: After reading the paper several times I still don't see the significant novelty of this paper. (A) The paper converts each hidden random variable to a deterministic variable via the reparameterization trick. This is a well-known technique. The VAE, for example,  uses this technique during training. Although the paper argues that "we do not primarily use reparameterization trick for model training. Rather, we leverage it to convert the latent variables z in DGMs to deterministic variables so that we can effectively distill knowledge from a compact form of DGM", but isn't this very straightforward? I don't see any big difference between using the reparameterization trick during training and KD. The authors should provide a discussion on this. (B) I don't see a difference between equations (4) and (2) when applying to VAE because, during VAE training, the sampling over the auxiliary random variables \epsilon is implicitly included even though we just apply equation (2). (3) Equations (5) and (6) look very intuitive and straightforward. I am more interested in knowing what theoretical guarantee we can have when using these losses. (C) For experimental evaluation, could you compare your model with more state-of-the-art KD baselines? (eg., Figures 4 and 5 and table 1).

**Summary Of The Paper:**

This paper proposes to use the reparameterization trick to convert the latent variables in DGMs to deterministic variables in the context of KD. It then proposes a surrogate distillation loss and latent distillation loss and evaluates the performance of the proposed method in three applications. Experimental results confirm the effectiveness of the proposed model.

**Summary Of The Review:**

I am mainly concerned about the novelty and clarity of this paper. At the current stage, I don't recommend the paper for acceptance.

---

> ### Author Response · Authors · 2021-11-21
> **Response to Reviewer 3  (Part 1/2)**
>
> **Q3.1**: On the significant novelty of this paper and the big difference between using the reparameterization trick during training and KD. The authors should provide a discussion on this.
>
> **A3.1**: We would like to point out that the definition of reparameterization trick in our framework is different from the classical one for VAE training [Kingma 2013]. Our reparameterization is more general than that for propagating gradients for model training. Specifically, reparameterization in our work is used for compacting the DGMs to semi-auxiliary forms rather than propagating gradients. Different from the classical reparameterization for model training that requires continuous latent variables, ours can be applied to a wider range of variables $\boldsymbol{z}$, including both continuous and discrete variables. In addition, the transformation function $g(\cdot)$ in our framework can be either differentiable or non-differentiable. Hence, our method can be applied to much more DGMs than the classical one. Table 6 summarizes the different requirements for our method and the classical reparameterization trick. Besides, we discuss how to apply our method to DGMs with discrete latent variables in our revised manuscript. We further conduct experiments on Helmholtz Machine (HM) model with discrete variables to demonstrate the good performance of our method, as introduced in Section 4.1.
>
> Table 6: Different requirements for our method and the classical reparameterization trick.
>
> | Requirements                       | Traditional reparameterization | Our framework                       |
> |------------------------------------|--------------------------------|-------------------------------------|
> | Transformation function $g(\cdot)$ | differentiable                 | differentiable / non-differentiable |
> | Latent variable $\boldsymbol{z}$   | continuous                     | continuous / discrete               |
> |    |                |              |
>
> Based on the above discussions, we would like to emphasize the main contributions of this work below.
> - To the best of our knowledge, this work is the first to develop a novel unified knowledge distillation (KD) framework for deep DGMs, especially for those with multiple layers of continuous/discrete latent variables.
> - We offer a theoretical proof that the proposed surrogate distillation loss is an upper bound of that of the regular KD.
> - We also present the subtle connections between our method and some existing KD approaches.
> - We conduct extensive experiments on different kinds of DGMs, such as hierarchical VAE, complicated VRNN, and HM with discrete latent variables, to evaluate the performance of the proposed framework. Experimental results on multiple benchmark datasets show that our approach can not only achieve high accuracy for data-free deep DGMs compression but also improve the continual learning performance of VAEs based on knowledge distillation.
>
>
> [Kingma 2013] Kingma, Diederik P., and Max Welling. "Auto-encoding variational bayes." arXiv preprint arXiv:1312.6114 (2013).

---

> > ### Author Response · Authors · 2021-11-21
> > **Response to Reviewer 3 (Part 2/2)**
> >
> > **Q3.2:** On the difference between equations (4) and (2) when applying to VAE, because the sampling over the auxiliary random variables $\boldsymbol{\epsilon}$ is implicitly during VAE training. I don't see a difference between equations (4) and (2) when applying to VAE because, during VAE training, the sampling over the auxiliary random variables $\boldsymbol{\epsilon}$ is implicitly included even though we just apply equation (2). (3). Equations (5) and (6) look very intuitive and straightforward.
> >
> > **A3.2:** For VAE distillation, we would like to clarify that Equation (4) and Equation (2) are completely different. Equation (2) $\mathcal{L} _{kd} = \mathbb{E} _{p _{data}(\boldsymbol{x})}\left [d(p_\phi(\boldsymbol{y} | \boldsymbol{x}), p_\theta(\boldsymbol{y} | \boldsymbol{x})) \right]$ is a vanilla distillation loss. In VAE distillation, there is no input variable, so $\boldsymbol{x} = \emptyset$. Then Equation (2) is reduced to $\mathcal{L} _{kd} = d(p_\phi(\boldsymbol{y}), p_\theta(\boldsymbol{y})) $, which is *intractable* because we can not marginalize all latent variables. In order to minimize Equation (2), we propose Equation (4) $ \mathcal{L} _{\mathit{sd}} = \mathbb{E} _{p_\phi(\boldsymbol{\epsilon})p _{data}(\boldsymbol{x})}\left[ d(p_\phi(\boldsymbol{y}| \boldsymbol{\epsilon},\boldsymbol{x}), p_\theta(\boldsymbol{y}|\boldsymbol{\epsilon},\boldsymbol{x}) ) \right]$, which is an upper bound of Equation (2), as proved in Appendix B. With $\boldsymbol{x} = \emptyset$, Equation (4) can be reduced to $ \mathcal{L} _{\mathit{sd}} = \mathbb{E} _{p_\phi(\boldsymbol{\epsilon})}\left[ d(p_\phi(\boldsymbol{y}| \boldsymbol{\epsilon}), p_\theta(\boldsymbol{y}|\boldsymbol{\epsilon})) \right]$. It is tractable because $p(\boldsymbol{y} | \boldsymbol{\epsilon}) $ can be computed by function composition. We take a 5-layer hierarchical VAE in our experiment as example, as illustrated in Fig. 6 (a). We can compute $p(\boldsymbol{y} | \boldsymbol{\epsilon}) $ by sequentially calculating $\boldsymbol{z}_2 = g(\boldsymbol{z}_1, \boldsymbol{\epsilon}_2)$, then $ \boldsymbol{z}_3 = g(\boldsymbol{z}_2, \boldsymbol{\epsilon}_3),\boldsymbol{z}_4 = g(\boldsymbol{z}_3, \boldsymbol{\epsilon}_4),\boldsymbol{z}_5 = g(\boldsymbol{z}_4, \boldsymbol{\epsilon}_5)$ and finally $  p(\boldsymbol{y}|\boldsymbol{\epsilon}) = p(\boldsymbol{y}|\boldsymbol{z}_5)$.
> >
> >
> > **Q3.3**: On the theoretical guarantee of the proposed losses
> >
> > **A3.3:** We have offered a theoretical proof that the surrogate distillation loss in Eq. (4) is an upper bound of the distillation loss in Eq.(2) when the dissimilarity measure is chosen to be KL divergence. Please refer to Proposition 3.1 for more details.
> >
> >
> > **Q3.4**:On comparing our model with more state-of-the-art KD baselines? (eg., Figures 4 and 5 and table 1).
> >
> > **A3.4**: To our best knowledge, this work is the first to develop a novel knowledge distillation framework for deep DGMs, especially for those with multiple layers of latent variables. Thus, it is very difficult to find more baselines for the tasks in Sections 4.1 and 4.3. Please note that in Section 4.1, we have already compared our *data-free* method with the commonly used baseline model trained from scratch[Lopes 2017, Yin 2020, Aguinaldo 2019].
> >
> > **References**
> >
> > [Lopes 2017]: Lopes R G, Fenu S, Starner T. Data-free knowledge distillation for deep neural networks[J]. arXiv preprint arXiv:1710.07535, 2017.
> >
> > [Yin 2020]: Yin, Hongxu, Pavlo Molchanov, Jose M. Alvarez, Zhizhong Li, Arun Mallya, Derek Hoiem, Niraj K. Jha, and Jan Kautz. "Dreaming to distill: Data-free knowledge transfer via deepinversion." In Proceedings of the IEEE/CVF Conference on Computer Vision and Pattern Recognition, pp. 8715-8724. 2020.
> >
> > [Aguinaldo 2019]: Aguinaldo A, Chiang P Y, Gain A, et al. Compressing gans using knowledge distillation[J]. arXiv preprint arXiv:1902.00159, 2019.

---

### Official Review · Reviewer_XcHW · 2021-11-03

**Correctness:** 2
**Technical Novelty And Significance:** 2
**Empirical Novelty And Significance:** 2
**Recommendation:** 5
**Confidence:** 4

**Main Review:**

# Writing:
* The paper is relatively readable. The main idea came across clear, but it can enjoy thorough editing, definitely in the experiment section.

# Method:
* The paper's main claim is that the local method can result in the accumulation of errors. However, the second term in Eq 7 somewhat does the same. In fact, one can view the proposed method as a combination of the marginalization and local methods together.

* I am not convinced by the idea of the semi-auxiliary graph that yields the loss function in Eq 3,4. The example in Fig 1 is too simple. For example, when we factor out the z's, the resulting graph on the observed variables is no longer DAG. Consider,  $y_1$ <-- z --> $y_2$ if $z$ is factored out then y_1 and y_2 are connected with undirected edge. How does Eq 3 handle such a case?

* The advantage of the method is mostly for DGMs with continuous latent variables. For discrete latent variables, the local distillation is used. I'm not sure how well this scales to models with the large number of discrete latent variables.

* One limitation is that the teacher and students should have the same architecture. Can it be extended to cases when the architectures are not necessarily the same?


# Experiment:
* The results in the experiment section are not convincing. For example, the difference between the performances of different methods is within the same standard error, especially for the local method and the proposed method.

* The choice of some of the tasks is questionable for the knowledge of DGM distillation. For example, why is continual learning a good task to show that the knowledge distillation method working? The argument made in the paper is that the Delta of forgetting the previous task is small, which is acceptable for continual learning; however, it is not clear why continual learning is a good task to show knowledge distillation.

* In figure 5: it is not clear why the authors claim that the proposed method produces better results. For example, why (c) is better than (e)? This experiment is very qualitative and subjective.

* None of the experiments (except HM) really motivate the knowledge distillation for DGM. VAE is a very simple DGM, and there is no real structure in the graphical model. The authors could have simulated data from a hierarchical graphical model and used a complex teacher to learn that model, then applied their method to show their approach can recover the true model from the teacher. For example, see [1] for examples such simulation -- a simple mixture model.

* I strongly recommend the authors look into the metrics introduced in [2]. Several metrics and experiments presented in that paper can be adopted or adapted for the DGM knowledge distillation.

# References

[1] https://arxiv.org/pdf/1603.06277.pdf
[2] https://arxiv.org/pdf/2106.05945.pdf

**Summary Of The Paper:**

The authors propose a method for knowledge distillation specifically for  Deep Directed Graphical Models. The authors compare their method with marginalization methods, which integrates the latent variables out, and the factorized (local) method, which distills knowledge between teacher and student on each factor. They validate their approach on continual learning, model compression, and discriminative learning.



**Summary Of The Review:**

* The paper is relatively clear.

* The experiments are not well-chosen and the results are not convincing.

* The method section does not explore the idea of knowledge distillation for DGM deeply. There are questions about the generalizability and scalability of the proposed method.

---

> ### Author Response · Authors · 2021-11-21
> **Response to Reviewer 2 (Part 1/4)**
>
> **Q 2.1**: On the proposed method can be viewed as a combination of the marginalization and local methods together.
>
> **A 2.1**:  We would like to point out that the proposed method is not a combination of the marginalization and local methods together. Specifically, the marginalized distillation is generally intractable, as discussed in Appendix A.1. Our method, however, can overcome the intractability issue of marginalized distillation because $p(\boldsymbol{y} | \boldsymbol{\epsilon}, \boldsymbol{x}) $ can be factorized to $ \prod_i p(\boldsymbol{y}_i | \boldsymbol{\epsilon}_\mathit{\leq i}, \boldsymbol{y}_\mathit{< i},\boldsymbol{x})$.
>
> Furthermore, we would like to elaborate the difference between the local distillation and our latent distillation. Specifically, if latent variables are continuous, different from local distillation, our latent distillation loss in Equation (6) serves to assist in mitigating the gradient vanishing problem in a very deep DGM. In our work, $\lambda$ in Equation (7) is a very *small* value. Even though $\lambda$ is set to 0, our method still works very well, as illustrated in Appendix G.2. If latent variables are discrete, our method still differs from local distillation. For example, if there is an imitation error between $p_\theta(\boldsymbol{z}_\mathit{i-1}| \boldsymbol{x}) $ and  $ p_\phi(\boldsymbol{z}_\mathit{i-1}| \boldsymbol{x})$, our method can mitigate this imitation error during the learning of $p_\theta(\boldsymbol{z}_i| \boldsymbol{z}_\mathit{i-1},\boldsymbol{x}) $. This is because when distilling latent variable $\boldsymbol{z}_i$, Equation (6) solely keeps the $i$-th layer of latent variables unchanged while converting all the previous $i-1$ layers of latent variables into an auxiliary form. Namely, we only share $\boldsymbol{\epsilon}_\mathit{<i}$ between teacher and student rather than $\boldsymbol{z}_\mathit{i-1}$. For local distillation, however, $p_\theta(\boldsymbol{z}_i| \boldsymbol{z}_\mathit{i-1},\boldsymbol{x})$ is learned by simply minimizing $\mathbb{E}_\mathit{p_\mathit{data}(\boldsymbol{x})p_\phi(\boldsymbol{z}_\mathit{i-1}| \boldsymbol{x})} \left[d( p_\phi(\boldsymbol{z}_i| \boldsymbol{z}_\mathit{i-1},\boldsymbol{x}) , p_\theta(\boldsymbol{z}_i| \boldsymbol{z}_\mathit{i-1},\boldsymbol{x}) ) \right]$, imitation error between $p_\theta(\boldsymbol{z}_\mathit{i-1}| \boldsymbol{x}) $ and  $ p_\phi(\boldsymbol{z}_\mathit{i-1}| \boldsymbol{x})$ can hardly be mitigated because $\boldsymbol{z}_\mathit{i-1}$ is directly shared between teacher and student.
>
>
>
> **Q 2.2:** On not convinced by the idea of the semi-auxiliary graph that yields the loss function in Eq 3, 4. The example in Fig 1 is too simple. when we factor out the $z$'s, the resulting graph on the observed variables is no longer DAG Consider, $\boldsymbol{y}_1 \leftarrow \boldsymbol{z} \rightarrow \boldsymbol{y}_2$ if $\boldsymbol{z}$ is factored out then $\boldsymbol{y}_1$ and $\boldsymbol{y}_2$ are connected with undirected edge. How does Eq. 3 handle such a case?
>
>
> **A 2.2:** We would like to point out that we do not "factorize out $\boldsymbol{z}$" in our work, since we do not integrate latent variable $\boldsymbol{z}$ in our method. Instead, we convert $\boldsymbol{z}$ to its auxiliary form in our framework using reparameterization trick. Therefore, the DGM example suggested by the Reviewer can be converted to $\boldsymbol{\epsilon}$ -> (deterministic) $\boldsymbol{z}$ -> $\boldsymbol{y}_1$ \& $\boldsymbol{y}_2$. It can still be seen as a DGM.
>
> Importantly, our method can deal with more complex DGMs than that suggested by the Reviewer. Fig. 6 (c) in the appendix shows an example of a complicated DGM, called VRNN, that contains many latent variables and target variables. We have conducted experiments to evaluate the VRNN model in Section 4.1. Evaluation results on two benchmark datasets demonstrate that the proposed KD method can handle complicated DGMs, and it outperforms the other baselines. For more details, please refer to Section 4.1.

---

> > ### Author Response · Authors · 2021-11-21
> > **Response to Reviewer 2 (Part 2/4)**
> >
> > **Q 2.3:** On scale to models with a large number of discrete latent variables.
> >
> > **A 2.3:** Our proposed method can also be scalable to a DGM with a large number of discrete latent variables. We take a DGM with $L$ (e.g, 20) layers of discrete latent variables as an example to demonstrate the good performance of our approach, as shown in Fig. 11 in the appendix. In our experiment, we let students mimic the output distribution of the teacher with $L$-layers latent variables for $L \in 1,\dots,20$. Each layer of the teacher follows the binomial distribution $p(\boldsymbol{z}_{i+1}|\boldsymbol{z}_i) = \mathcal{B}(1000, \boldsymbol{p}(\boldsymbol{z}_i))$, where $\boldsymbol{p}(\boldsymbol{z}_i) = (\frac{\boldsymbol{z}_i}{1000})^{1.2}$; $p(\boldsymbol{z}_1)$ is a uniform distribution $U[0, 1]$. The student is parameterized by neural networks with a residual structure [He 2016]. Then *density ratio estimation* [Nguyen 2010, Sugiyama 2012] is used to measure the KL divergence between the teacher and the student. We can observe from Fig. 11 in the appendix that even if all the latent variables are discrete, our method still has the capability of mitigating the imitation error occurred in the shallow layers and achieves a very good imitation performance. For the local distillation method, however, its imitation errors are accumulated as the number of layers increases.
> >
> > **References:**
> >
> > [Nguyen 2010] Nguyen, X., Wainwright, M. J., and Jordan, M. I. (2010). Estimating divergence functionals and the likelihood ratio by convex risk minimization. IEEE Transactions on Information Theory, 56(11), 5847-5861.
> >
> > [Sugiyama 2012] Sugiyama, M., Suzuki, T., \& Kanamori, T. (2012). Density-ratio matching under the Bregman divergence: a unified framework of density-ratio estimation. Annals of the Institute of Statistical Mathematics, 64(5), 1009-1044.
> >
> > [He 2016] He, Kaiming, Xiangyu Zhang, Shaoqing Ren, and Jian Sun. "Deep residual learning for image recognition." In Proceedings of the IEEE conference on computer vision and pattern recognition, pp. 770-778. 2016.
> >
> > **Q2.4**: On limitation about the teacher and students should have the same architecture. Can it be extended to cases when the architectures are not necessarily the same?
> >
> > **A2.4:** In our method, teacher and students do not have to share the same architecture. Nevertheless, we admit that there is a limitation about the architectures of the teacher and student. They need to consist of the same number of random variables (these variables can be arranged in different structures). We have discussed the limitation of our method in the appendix in our revised version. For future work, we plan to extend the proposed method to deal with more difficult cases where the teacher and students have different numbers of random variables.
> >
> > **Q 2.5:** On why continual learning is a good task to show knowledge distillation
> >
> > **A2.5:** According to the prior works [Yin 2020, Zhai 2019, Zhou 2019, Ramapuram 2020, Li 2017, Dhar 2019, Rebuffi 2017] , continual learning is one of the most important applications for knowledge distillation. It is a representative task for evaluating the performance of data-free knowledge distillation. Continual learning can benefit from transferring knowledge from the old model to a new model without access to the original dataset, which is aligned with the definition of data-free knowledge distillation (KD). That is why we apply our KD method to continual learning.
> >
> > **References**
> >
> > [Yin 2020] Yin, Hongxu, Pavlo Molchanov, Jose M. Alvarez, Zhizhong Li, Arun Mallya, Derek Hoiem, Niraj K. Jha, and Jan Kautz. "Dreaming to distill: Data-free knowledge transfer via deepinversion." In Proceedings of the IEEE/CVF Conference on Computer Vision and Pattern Recognition, pp. 8715-8724. 2020.
> >
> > [Zhai 2019] Zhai, Mengyao, Lei Chen, Frederick Tung, Jiawei He, Megha Nawhal, and Greg Mori. "Lifelong gan: Continual learning for conditional image generation." In Proceedings of the IEEE/CVF International Conference on Computer Vision, pp. 2759-2768. 2019.
> >
> > [Zhou 2019] Zhou, Peng, Long Mai, Jianming Zhang, Ning Xu, Zuxuan Wu, and Larry S. Davis. "M2kd: Multi-model and multi-level knowledge distillation for incremental learning." arXiv preprint arXiv:1904.01769 (2019).
> >
> > [Ramapuram 2020] Ramapuram, Jason, Magda Gregorova, and Alexandros Kalousis. "Lifelong generative modeling." Neurocomputing 404 (2020): 381-400.
> >
> > [Li 2017] Li, Zhizhong, and Derek Hoiem. "Learning without forgetting." IEEE transactions on pattern analysis and machine intelligence 40, no. 12 (2017): 2935-2947.
> >
> > [Dhar 2019] Dhar, Prithviraj, Rajat Vikram Singh, Kuan-Chuan Peng, Ziyan Wu, and Rama Chellappa. "Learning without memorizing." In Proceedings of the IEEE/CVF CVPR, pp. 5138-5146. 2019.
> >
> > [Rebuffi 2017] Rebuffi, Sylvestre-Alvise, Alexander Kolesnikov, Georg Sperl, and Christoph H. Lampert. "icarl: Incremental classifier and representation learning." In Proceedings of the CVPR, pp. 2001-2010. 2017.

---

> > > ### Author Response · Authors · 2021-11-21
> > > **Response to Reviewer 2 (Part 3/4)**
> > >
> > > **Q 2.6**:  On the experimental result is very qualitative and subjective in Fig.5. Why (c) is better than (e)?}
> > >
> > > **A 2.6**: To the best of our knowledge, there do not exist any suitable and well-known quantitative metrics on this dataset. Currently, qualitative comparison is still a commonly used way to measure the performance [Graves 2013, Chung 2015]. We can observe from Fig. 5 (c) that the strokes generated by our method are as clear and readable as those of the teacher in Fig.5 (b). However, the strokes in Fig.5 (e) are very small and very unreadable. That is why (c) is better than (e) in Fig. 5. For future work, we plan to evaluate the proposed framework on the other tasks in order to show its good performance.
> > >
> > > **References**
> > >
> > > [Graves 2013] Graves, Alex. "Generating sequences with recurrent neural networks." arXiv preprint arXiv:1308.0850 (2013).
> > >
> > > [Chung 2015] Chung, J., Kastner, K., Dinh, L., Goel, K., Courville, A. C., \& Bengio, Y. (2015). A recurrent latent variable model for sequential data. Advances in neural information processing systems, 28, 2980-2988.
> > >
> > > **Q 2.7**: None of the experiments (except HM) really motivate the knowledge distillation for DGM. Do experiments on complex VAE models.
> > >
> > > **A 2.7**: We would like to note that we already conducted experiments on some complicated VAE models in our submission, as illustrated in Fig. 6 in the appendix. Please note that the VAE model in our experiment refers to a hierarchical VAE consisting of 5-layer random variables rather than a simple one-layer VAE. The hierarchical VAE architecture is illustrated in Fig 6(a) in the appendix. Note that the model presented in Section 2.1 in literature [Johnson 2016] suggested by the Reviewer is actually a simpler case of our hierarchical VAE in Section 4.1.
> > >
> > > Besides, we have conducted experiments on the more complicated variational recurrent neural networks (VRNN), as shown in Fig. 6(c). The model presented in Section 2.2 in [Johnson 2016] suggested by the reviewer is a simpler case of VRNN. The experimental results demonstrate the good performance of our method on complex VAE models. For more details, please refer to Section 4.1 in our revised version.
> > >
> > > **References**
> > >
> > > [Johnson 2016] Johnson, M. J., Duvenaud, D. K., Wiltschko, A., Adams, R. P., \& Datta, S. R. (2016). Composing graphical models with neural networks for structured representations and fast inference. Advances in neural information processing systems, 29, 2946-2954.

---

> > > > ### Author Response · Authors · 2021-11-21
> > > > **Response to Reviewer 2 (Part 4/4)**
> > > >
> > > >
> > > > **Q 2.8:** On more metrics to evaluate the proposed knowledge distillation
> > > >
> > > > **A 2.8:** Thanks for the suggestions. We have used another two metrics, Calibration Error (ECE) [Guo 2017] and negative log likelihood (NLL), from the literature [Stanton 2021] suggested by the Reviewer to evaluate the performance of our method. The experimental results on two benchmark datasets are shown in the following Tables 4 and 5. We can observe that our method consistently achieves the best NLL on the two datasets. Besides, our method has lower ECE than all the baselines except for Dropout method. We have added these experimental results in Section 4.3 our revised manuscript.
> > > >
> > > > Table 4: Distillation metrics of different methods averaged over 5 random seeds on Fashion MNIST dataset.
> > > >
> > > > |           |     \#Parameters    |         Accuracy        |             NLL            |             ECE            |
> > > > |-----------|:-------------------:|:-----------------------:|:--------------------------:|:--------------------------:|
> > > > | Teacher   |        2.60M        |          90.40          |           0.4657           |           0.0592           |
> > > > | Vanilla   | 0.40M (6.5$\times$) |      89.69$\pm$0.12     |      0.6512$\pm$0.0051     |      0.0768$\pm$0.0011     |
> > > > | Dropout   | 0.40M (6.5$\times$) |      89.69$\pm$0.13     |      0.6527$\pm$0.0089     |      0.0781$\pm$0.0014     |
> > > > | VIB       | 0.40M (6.5$\times$) |      89.70$\pm$0.24     |      0.4799$\pm$0.0122     |      0.0640$\pm$0.0033     |
> > > > | MC-VIB    | 0.40M (6.5$\times$) |      90.35$\pm$0.08     |      0.4589$\pm$0.0016     |      0.0646$\pm$0.0010     |
> > > > | Local-VIB | 0.40M (6.5$\times$) |      90.38$\pm$0.06     |      0.4667$\pm$0.0011     |      0.0593$\pm$0.0001     |
> > > > | Our-VIB   | 0.40M (6.5$\times$) | **90.43$\pm$0.12** | **0.4291$\pm$0.0055** | **0.0576$\pm$0.0012** |
> > > > |      |               |              |             |               |
> > > >
> > > >
> > > > Table 5: Distillation metrics of different methods averaged over 5 random seeds on SVHN dataset.
> > > >
> > > > |           |     \#Parameters    |         Accuracy        |             NLL            |             ECE            |
> > > > |-----------|:-------------------:|:-----------------------:|:--------------------------:|:--------------------------:|
> > > > | Teacher   |        2.60M        |          83.97          |           0.6589           |           0.0484           |
> > > > | Vanilla   | 0.40M (6.5$\times$) |      81.07$\pm$1.68     |      0.7187$\pm$0.0356     |      0.0294$\pm$0.0090     |
> > > > | Dropout   | 0.40M (6.5$\times$) |      76.46$\pm$4.95     |      0.8306$\pm$0.1185     | **0.0229$\pm$0.0050** |
> > > > | VIB       | 0.40M (6.5$\times$) |      83.01$\pm$0.36     |      0.6454$\pm$0.0068     |      0.0244$\pm$0.0015     |
> > > > | MC-VIB    | 0.40M (6.5$\times$) |      83.77$\pm$0.32     |      0.6893$\pm$0.0229     |      0.0494$\pm$0.0017     |
> > > > | Local-VIB | 0.40M (6.5$\times$) |      83.94$\pm$0.01     |      0.6588$\pm$0.0003     |      0.0476$\pm$0.0003     |
> > > > | Our-VIB   | 0.40M (6.5$\times$) | **84.04$\pm$0.09** | **0.6307$\pm$0.0082** |      0.0346$\pm$0.0006     |
> > > > |      |               |              |             |               |
> > > >
> > > > **References**
> > > >
> > > > [Guo 2017] Guo, Chuan, Geoff Pleiss, Yu Sun, and Kilian Q. Weinberger. "On calibration of modern neural networks." In International Conference on Machine Learning, pp. 1321-1330. PMLR, 2017.
> > > >
> > > > [Stanton 2021] Stanton, S., Izmailov, P., Kirichenko, P., Alemi, A. A., \& Wilson, A. G. (2021). Does Knowledge Distillation Really Work?. arXiv preprint arXiv:2106.05945.

---

### Official Review · Reviewer_sRSN · 2021-11-04

**Correctness:** 3
**Technical Novelty And Significance:** 2
**Empirical Novelty And Significance:** 2
**Recommendation:** 5
**Confidence:** 4

**Main Review:**

I think the proposed distillation framework is novel, and the entire paper is well-organized. However, I have the following concerns/questions,

1. The title claims for a unified KD framework for deep DGMs. However, the proposed distillation loss only applies to DGMs where the latent variable z has a reparameterization form. I believe the author should highlight this limitation of their framework.


2. In equation (3), the expected KL divergence is computed over $p_{\phi}(y_{<j}|x)$.  I am wondering if the following three ways will improve the performance of local distillation? Though they are no longer equivalent to equation (2), my intuition is that, for the original objective (equation (3)), the conditional KL divergence is computed with respect to the teacher's distribution $p_{\phi}(y_{<j}|x)$, however, at inference time,  the conditional distribution of the student model is computed based on its own distribution $p_{\theta}(y_{ <j}|x)$. This makes the training objective does not consistent with the test objective.


- $
\mathcal{L}\_{kd} = \mathbb{E}\_{p_{data}(x)}\left[\mathbb{E}\_{p_{\theta}(y\_{<j}|x)} [KL(p_{\phi}(y_j|y_{<j}, x)|| p_{\theta}(y_j|y_{<j}, x) )] \right]
$


- $\mathcal{L}\_{kd} = \mathbb{E}\_{p_{data}(x)} \left[
\mathbb{E}\_{p_{\theta}(y_{<j}|x)} [KL( p_{\theta}(y_j|y_{<j}, x) || p_{\phi}(y_j|y_{<j}, x)) ]
 \right]$

- $\mathcal{L}\_{kd} = \mathbb{E}\_{p_{data}(x)} \left[\sum_j (1-\lambda)\cdot \mathbb{E}\_{p_{\phi}(y_{<j}|x)} [KL(p_{\phi}(y_j|y_{<j}, x)|| p_{\theta}(y_j|y_{<j}, x) )]   +
\lambda\cdot\mathbb{E}\_{p_{\theta}(y_{<j}|x)} [KL(p_{\phi}(y_j|y_{<j}, x)|| p_{\theta}(y_j|y_{<j}, x) )]
 \right]$  (for some $\lambda \in [0, 1]$)

3. Can you explain more on why do you remove $ \epsilon_i$ in equation (6)? I don't understand why such a change will better penalize the dissimilarity of latent variables $z_i$?

4. Can you provide some qualitative comparisons between the samples from the distilled DGMs on the CelebA dataset?

**Summary Of The Paper:**

This paper proposes a new knowledge distillation framework for directed graphical models based on the reparameterization trick. The new distillation framework overcomes the intractable marginals in marginalized distillation, and error accumulation in local distillation. Empirically, the new distillation framework surpasses the baselines on deep generative model compression, VAE continual learning, and discriminative DGMs compression.

**Summary Of The Review:**

Overall, I like the solution proposed by this paper to address the issues in marginalized distillation and local distillation. However, I believe the authors did not fully investigate the failure reasons for local distillation. The argument in the current paper is kind of vague and not well-supported. I believe a more rigorous analysis and carefully designed experiments are needed to illustrate the argument (e.g., as I mentioned above). I will increase my rating if the authors can provide more convincing results.

---

> ### Author Response · Authors · 2021-11-21
> **Response to Reviewer 1 (Part 1/3)**
>
> **Q1.1**: On highlighting the limitation of the proposed framework that only applies to DGMs where the latent variable $\boldsymbol{z}$ has a reparameterization form.
>
> **A1.1**: We have discussed the limitation of the proposed framework in Appendix J in our revised version. Our framework is applied to DGMs where latent variables can be reparameterized. However, we would like to point out that the definition of the reparameterization trick in our framework is slightly different from the classical one for VAE training [Kingma 2013]. Our reparameterization is more general than that for propagating gradients for model training. Specifically, reparameterization in our work is used for compacting the DGMs to semi-auxiliary forms rather than propagating gradients. Different from the classical reparameterization for model training that requires continuous latent variables, ours can be applied to both continuous and discrete variables. In addition, the transformation function $g(\cdot)$ in our framework can be either differentiable or non-differentiable. Hence, our method can be applied to much more DGMs than the classical one.
>
> The following Table 1 summarizes the different requirements for our method and the classical reparameterization trick. Besides, we discuss how to apply our method to DGMs with discrete latent variables in our revised manuscript. We further conduct experiments on Helmholtz Machine (HM) model with discrete variables to demonstrate the good performance of our method, as introduced in Section 4.1.
>
> Table 1: Different requirements for our method and the classical reparameterization trick.
>
> | Requirements                       | Traditional reparameterization | Our framework                       |
> |------------------------------------|--------------------------------|-------------------------------------|
> | Transformation function $g(\cdot)$ | differentiable                 | differentiable / non-differentiable |
> | Latent variable $\boldsymbol{z}$   | continuous                     | continuous / discrete               |
> |                                       |                    |                                              |
>
> [Kingma 2013] Kingma, Diederik P., and Max Welling. "Auto-encoding variational bayes." arXiv preprint arXiv:1312.6114 (2013).

---

> > ### Author Response · Authors · 2021-11-21
> > **Response to Reviewer 1 (Part 2/3)**
> >
> > **Q1.2:** On three ways will improve the performance of local distillation.
> >
> > **A1.2:**  We have conducted experiments to evaluate the performance of local distillation based on three ways suggested by the Reviewer. We take data-free hierarchical VAE compression as an example. In our experiment, we try to achieve $15 \times$ compression of the VAE models on the CelebA dataset. We add the *student generated* local distillation loss suggested by the Reviewer into the objective function. We choose different weights, $\lambda \in \{0.8, 0.5, 0.2, 0.1, 0.01\}$, on the latent distillation loss in the experiment. In addition, KL divergence is chosen to be either $KL\left ( p_\phi(\boldsymbol{y}_j|\mathit{Pa}(\boldsymbol{y}_j),\boldsymbol{x})\parallel p_\theta(\boldsymbol{y}_j|\mathit{Pa}(\boldsymbol{y}_j),\boldsymbol{x}) \right )$ or $KL\left (p_\theta(\boldsymbol{y}_j|\mathit{Pa}(\boldsymbol{y}_j),\boldsymbol{x}) \parallel p_\phi(\boldsymbol{y}_j|\mathit{Pa}(\boldsymbol{y}_j),\boldsymbol{x})\right)$.
> >
> > Unfortunately, for different KL divergence and $\lambda$, the suggested models fail to converge such that they only generate very low-quality images. Table 2 illustrates the comparison of FID for three different local distillation methods. We can observe that their best FID score is 243.8, which is much higher than 90 in our method. We also present the results of different combinations of $\lambda$ and KL divergence in Table 3 below.
> >
> > The above experimental results further verify our analysis on the failure reason of local distillation in the manuscript. Sometimes, $p_\theta(\mathit{Pa}(\boldsymbol{y}_j)|\boldsymbol{x})$ may be entirely different from $p_\phi(\mathit{Pa}(\boldsymbol{y}_j)|\boldsymbol{x})$. Hence, for local distillation, when we learn $p_\theta(\boldsymbol{y}_j|\mathit{Pa}(\boldsymbol{y}_j),\boldsymbol{x})$, there will be a noticeable mismatch between its training set $p_\phi(\mathit{Pa}(\boldsymbol{y}_j)|\boldsymbol{x})$ and testing set $p_\theta(\mathit{Pa}(\boldsymbol{y}_j)|\boldsymbol{x})$, leading to the inferior performance of $p_\theta(\boldsymbol{y}_j|\mathit{Pa}(\boldsymbol{y}_j),\boldsymbol{x})$. The inferior performance of $p_\theta(\boldsymbol{y}_j|\mathit{Pa}(\boldsymbol{y}_j),\boldsymbol{x})$ together with the mismatch between $p_\phi(\mathit{Pa}(\boldsymbol{y}_j)|\boldsymbol{x})$ and $p_\theta(\mathit{Pa}(\boldsymbol{y}_j)|\boldsymbol{x})$ will further enlarge the mismatch between $p_\phi(\boldsymbol{y}_j|\boldsymbol{x})$ and $p_\theta(\boldsymbol{y}_j|\boldsymbol{x})$. As a result, it causes massive negative impact on the learning of the succeeding random variables $p_\theta(\boldsymbol{y}_\mathit{j+1}|\mathit{Pa}(\boldsymbol{y}_\mathit{j+1}),\boldsymbol{x})$. This vicious circle describes the detailed process of error accumulation.
> >
> > Table 2: Best FID of three methods suggested by the Reviewer in $15 \times$ VAE compression.
> >
> > | Methods suggested by Reviewer |  FID  |
> > |-------------------------------|:-----:|
> > | Local distillation loss 1     | 317.6 |
> > | Local distillation loss 2     | 243.8 |
> > | Local distillation loss 3     | 243.8 |
> > |                               |       |
> >
> > Table 3: FID of student models with different $\lambda$ and KL divergence in $15 \times$ VAE compression.
> >
> > | $\lambda$ | KL                      | FID   |
> > |-----------|-------------------------|-------|
> > | 1.0       | $KL(\phi \parallel \theta)$ | 317.6 |
> > | 1.0       | $KL(\theta \parallel \phi)$ | 243.8 |
> > | 0.8       | $KL(\phi \parallel \theta)$ | 311.3 |
> > | 0.8       | $KL(\theta \parallel \phi)$ | 320.6 |
> > | 0.5       | $KL(\phi \parallel \theta)$ | 314.4 |
> > | 0.5       | $KL(\theta \parallel \phi)$ | 312.2 |
> > | 0.2       | $KL(\phi \parallel \theta)$ | 322.6 |
> > | 0.2       | $KL(\theta \parallel \phi)$ | 323.1 |
> > | 0.1       | $KL(\phi \parallel \theta)$ | 326.5 |
> > | 0.1       | $KL(\theta \parallel \phi)$ | 297.8 |
> > | 0.01      | $KL(\phi \parallel \theta)$ | 349.3 |
> > | 0.01      | $KL(\theta \parallel \phi)$ | 258.3 |

---

> > > ### Author Response · Authors · 2021-11-21
> > > **Response to Reviewer 1 (Part 3/3)**
> > >
> > > **Comment:**
> > >
> > > **Q1.3**: On why removing $\boldsymbol{\epsilon}_i$ in Equation (6) and why such a change will better penalize the dissimilarity of latent variables.
> > >
> > > **A 1.3**: The primary reason of removing $\boldsymbol{\epsilon}_i$ in Eq. (6) is that $\boldsymbol{z}_i$ would be a deterministic variable if we keep it converted using reparameterization trick. This will severely limit our choice of $d(\cdot,\cdot)$ to penalize the dissimilarity between deterministic $p_\phi(\boldsymbol{z}_i| \boldsymbol{\epsilon}_\mathit{\leq i},\boldsymbol{x})$ and $p_\theta(\boldsymbol{z}_i| \boldsymbol{\epsilon}_\mathit{\leq i},\boldsymbol{x}) $. For instance, if we do not remove $\boldsymbol{\epsilon}_i$, $\boldsymbol{z}_i$ would be a deterministic variable, and $p_\phi(\boldsymbol{z}_i| \boldsymbol{\epsilon}_\mathit{\leq i},\boldsymbol{x})$, $\ p_\theta(\boldsymbol{z}_i| \boldsymbol{\epsilon}_\mathit{\leq i},\boldsymbol{x}) $ become two degenerate distributions. When $p_\phi(\boldsymbol{z}_i| \boldsymbol{\epsilon}_\mathit{\leq i},\boldsymbol{x})$ and $\ p_\theta(\boldsymbol{z}_i| \boldsymbol{\epsilon}_\mathit{\leq i},\boldsymbol{x}) $ are not exactly the same, they have disjoint support, causing their KL divergence to be infinity. In contrast, if we convert $\boldsymbol{z}_i$ back to its original form, e.g. Gaussian distribution, we have more choices of $d(\cdot,\cdot)$ to measure their dissimilarity. For instance, we can choose KL divergence, Jensen–Shannon divergence or Wasserstein distance to measure the dissimilarity based on different types of latent variables.
> > >
> > >
> > >
> > > **Q 1.4:** On qualitative comparisons between the samples from the distilled DGMs on the CelebA dataset}
> > >
> > > **A1.4**: We have conducted experiments to compare the generated samples by different knowledge distillation (KD) methods on the CelebA dataset, as illustrated in Fig. 8 in Appendix G.2. It can be observed from Fig. 8 (c) that images generated by our method are high-fidelity, as good as ground truth and those generated by the teacher, which means our method can imitate teacher's performance very well. Conversely, a limited-size VAE model is incapable of learning such a complicated distribution from scratch on its own. Hence, the images generated by the student VAE trained from scratch are low quality, as illustrated in Fig. 8 (e)

---

### Author Response · Authors · 2021-11-21
**General Response to Reviewers**

We would like to thank the reviewers for their thorough reviews and constructive comments on our manuscript. We have addressed the comments of individual reviewers below. In addition, we have uploaded a revised manuscript. Note that the key changes in our manuscript are highlighted in BLUE.

---

### Decision · Program_Chairs · 2022-01-20

**Decision:**

Reject

**Comment:**

The paper proposes a framework for distilling deep directed graphical models where the teacher and student models have the same number of latent variables z. The key idea is to reparameterize both models in terms of standardized random variables epsilon with fixed distributions and train the student to match the conditional distributions of the observed variables/targets given the values of the standardized RVs epsilon. The approach aims to avoid error compounding that affects the local distillation approach, where the student is trained to match conditional distributions of the teacher model (without the above reparameterization). To deal with discrete latent variables and vanishing gradients the authors augment the target matching loss with the latent distillation loss that matches the local distribution for each z_i given the standardized variables epsilon it depends on.

Positives
-The paper tackles an important problem.
-The idea of using reparameterization for distillation in this way makes a lot of sense for continuous latent variables and could be impactful.
-The experiments provide some evidence in support of the idea.

Negatives
-There are considerable issues with the clarity of writing: for example, it is really not clear how (and why) the method is supposed to work for discrete latent variables. The explanation provided by the authors in their response to the reviewers was helpful but still not clear enough.
-The fact that the teacher and student models need to have the same number of latent variables (and perhaps even the same structure) is a big limitation of the method given the claim of its generality, and thus needs to be clearly acknowledged and discussed. For example, the method cannot be used to train a student model with fewer latent variables than the teacher, which seems like a very common use case.
-The experimental evaluation is extensive but insufficient, in large part due to the evaluation metrics. Given that VAEs are trained by maximizing the ELBO (and distilled by minimizing a sum of KLs), it makes sense to also evaluate them based on the ELBO rather than solely on the FID, is done in the paper. The VRNN experiment would be much more informative if it included a quantitative evaluation (e.g. based on ELBO).

In summary, the paper has considerable potential but needs to be substantially improved before being published.